# SERPINB3 (SCCA1) inhibits cathepsin L and lysoptosis, protecting cervical cancer cells from chemoradiation

Songyan Wang[1], Cliff J. Luke [2,3], Stephen C. Pak [2], Victoria Shi[1], Liyun Chen [1], Jonathan Moore[1], Arlise P. Andress[1], Kay Jayachandran[1], Jin Zhang[1], Yi Huang [1], Marina Platik[1], Anthony A. Apicelli[1,3], Julie K. Schwarz[1,3], Perry W. Grigsby[1,3,4,5], Gary A. Silverman [2] & Stephanie Markovina [1,3✉]

The endogenous lysosomal cysteine protease inhibitor SERPINB3 (squamous cell carcinoma antigen 1, SCCA1) is elevated in patients with cervical cancer and other malignancies. High serum SERPINB3 is prognostic for recurrence and death following chemoradiation therapy. Cervical cancer cells genetically lacking SERPINB3 are more sensitive to ionizing radiation (IR), suggesting this protease inhibitor plays a role in therapeutic response. Here we demonstrate that SERPINB3-deficient cells have enhanced sensitivity to IR-induced cell death. Knock out of SERPINB3 sensitizes cells to a greater extent than cisplatin, the current standard of care. IR in SERPINB3 deficient cervical carcinoma cells induces predominantly necrotic cell death, with biochemical and cellular features of lysoptosis. Rescue with wild-type SERPINB3 or a reactive site loop mutant indicates that protease inhibitory activity is required to protect cervical tumor cells from radiation-induced death. Transcriptomics analysis of primary cervix tumor samples and genetic knock out demonstrates a role for the lysosomal protease cathepsin L in radiation-induced cell death in SERPINB3 knock-out cells. These data support targeting of SERPINB3 and lysoptosis to treat radioresistant cervical cancers.

[1] Department of Radiation Oncology, Washington University School of Medicine, St Louis, MO 63110, USA. [2] Department of Pediatrics, Washington University School of Medicine, St Louis, MO 63110, USA. [3] Alvin J Siteman Cancer Center, Washington University School of Medicine, St Louis, MO 63110, USA. [4] Division of Gynecologic Oncology, Department of Obstetrics and Gynecology, Washington University School of Medicine, St Louis, MO 63110, USA. [5] Division of Nuclear Medicine, Edward Mallinckrodt Institute of Radiology, Washington University School of Medicine, St Louis, MO 63110, USA. ✉email: smarkovina@wustl.edu

Cervical cancer remains a leading cause of cancer death for women worldwide[1]. Despite efforts to improve screening and human papilloma-virus (HPV) vaccination rates, overall mortality from the disease has not changed substantially in the last several decades[1,2]. Definitive chemoradiation is the standard of care for most women with cervical cancer, but is associated with recurrence rates as high as 30–70%[3,4], implying that many cervical cancers display inherent or adaptive cellular resistance to these therapies. For women who experience recurrent disease following chemoradiation therapy, less than 5% of women survive beyond five years[3]. Thus, there is an unmet need to understand of the molecular mechanisms of resistance to radiation and chemotherapy in cervical cancer, with the goal of developing novel therapies for both upfront and salvage therapy.

We previously demonstrated that elevated serum squamous cell carcinoma antigen (SCCA) is independently prognostic for recurrence and survival following definitive radiation therapy for cervical cancer[5,6]. Moreover, we found that failure of serum SCCA to normalize by the fourth week of treatment was an early indicator of failed response to therapy as indicated by positive post-therapy FDG-PET, and increased recurrence and death[6]. Clinical serum SCCA assays measure levels of SERPINB3 and SERPINB4 proteins, also known as SCCA1 and SCCA2, respectively. These proteins share high homology, and diverge in amino acid sequence primarily within the carboxy-terminal reactive site loop (RSL), which serves as a pseudo-substrate for specific proteases[7]. Binding and cleavage of the RSL by a target protease results in a rapid conformational change in the SERPIN, preventing further processive proteolysis of the SERPIN and results in a covalently bound SERPIN-protease complex that is ultimately degraded[8].

SERPINB3 is an intracellular cysteine protease inhibitor that is upregulated in many autoimmune diseases and cancer[5,9–14]. The *Caenorhabditis elegans* (C. elegans) homologue of SERPINB3, SRP-6, protects against lysosomal damage and organismal death by inhibiting cysteine protease activity induced by diverse stressors including hypotonic saline, hydrogen peroxide, DNA-damage and reactive oxygen species[15]. This phenotype can be rescued in the *srp-6* null animal by driving expression of wild-type SRP-6, but not RSL-mutant SRP-6, suggesting that inhibition of cysteine proteases is necessary for cytoprotection. SERPINB4 demonstrates specificity for chymotrypsin-like serine proteases in vitro[16], and has been shown to inhibit granzyme M-induced cell death when overexpressed in HeLa cells, but does not have other well-described cellular functions[17].

While high levels of SERPINB3 in the serum is associated with poor response to RT in patients with cervical cancer, it is not clear if SERPINB3 impacts radiation response on a cellular level. We hypothesized that SERPINB3 serves as a radioprotective factor in cervical cancer cells. Indeed we found that cervical cancer cell lines engineered by CRISPR-Cas9-mediated gene editing of the *SERPINB3* locus resulting in knockout (B3-KO) were significantly more sensitive to ionizing radiation in vitro compared to isogenic control cells, as determined by clonogenic cell survival assay[9]. The molecular mechanism of protection against radiation is unknown. Murakami et al found that transient transfection of vector-driven *SERPINB3* into 293 T human embryonic kidney cells resulted in higher survival as measured by MTT assay up to 36 h after treatment with radiation[18]. Examination of caspases important for apoptosis demonstrated decreased levels of caspase 9 cleavage and caspase-3 activity in cells transfected with SERPINB3. Based on these findings the authors concluded that SERPINB3 suppresses radiation-induced apoptosis. While the regulation of caspases is important in apoptotic cell death pathways, exogenous stressors such as IR often induce multiple cellular death pathways within a population of cells. Additionally,

radiation does not engage apoptosis effectively in most solid tumors due to common dysregulation of pro-apoptotic pathways[19]. Thus, the precise molecular mechanism of protection by SERPINB3 against radiation remains unknown.

In the current study, we found that the mechanism of SERPINB3-mediated radioprotection in cervical cancer cells is inhibition of cell death. Moreover, we identified the primary mode of cell death induced by radiation in SERPINB3-KO cells as lysoptosis-like, a distinct regulated lysosome-mediated necrosis pathway described in detail the companion publication (Luke et al, "Lysoptosis is an ancient and evolutionarily-conserved cell death pathway moderated by intracellular serpins," manuscript # COMMSBIO-20-2781-T). The central hallmark of lysoptosis is lysosomal membrane permeability with leakage of active cathepsin proteases, namely cathepsin L, into the cytosol, and is inhibited in the presence of SERPINB3 or pharmacologic cysteine protease inhibitors and selective cathepsin L inhibitors. Here we show that radiation induces lysoptosis-like necrotic cell death, with little to no evidence of apoptotic cell death, necroptosis, pyroptosis or ferroptosis. Additionally, the SERPINB3 RSL is required for protection against radiation, and knockout of the lysosomal cysteine protease cathepsin L (CTSL) partially protected SERPINB3-KO cervix tumor cells from radiation-induced death, further supporting the mechanism of lysosomal protease inhibition. To our knowledge, this is the first report that SERPINB3 protects cancer cells against radiation-induced necrosis, and does so via inhibition of the lysosomal cysteine protease cathepsin L. Moreover, we present evidence that ionizing radiation induces a lysosome-mediated cell death pathway in cancer cells, particularly when a protective factor is eliminated, suggesting a potential vulnerability for targeting radioresistant cancers.

## Results

**SERPINB3-KO cells are more sensitive to radiation and cisplatin-induced cell death.** For these studies, a panel of cervical cancer cell lines was selected to provide the most clinically relevant models. Two cell lines with high levels of SERPINB3 (SW756 and HT3) and two with low to undetectable levels of SERPINB3 (SiHa and C33A), had varying sensitivity to radiation as determined by clonogenic cell survival, with the cell lines expressing SERPINB3 more radioresistant than those without SERPINB3 (Supplemental Fig. 1a, b). Importantly, SW756 with a single integrated copy of the HPV type 18 and SiHa with an integrated copy of HPV type 16, were selected to represent commonly HPV-associated cancers. Conversely, HT3 and C33A cell lines, which do not have HPV DNA, were selected to represent the ~10% of HPV-negative cervical cancers (Supplemental Fig. 1a, b). Patients with HPV-negative cervical cancers in some clinical series have worse outcomes following chemoradiation[20], and in the case of head and neck squamous cell carcinoma, these cancers are more likely to be resistant to radiation treatment[21,22]. Thus, we felt it important to understand the impact of SERPINB3-KO on tumor cells of both HPV-positive and HPV-negative origins.

To determine if SERPINB3 protects cells against stress-induced death through a conserved mechanism similar to SRP-6, we first quantified cell death by exclusion of Sytox™ nucleic acid stain in cells treated with sham or increasing doses of radiation over a time course of several days. We found that clonal cell lines with CRISPR-Cas9-mediated SERPINB3-knockout (referred to as "B3-KO") in both HT3 and SW756 cervical cancer cells had significantly higher percent cell death at every dose and every timepoint following treatment with IR, compared to cell lines with the CRISPR-Cas9 vector containing no guide RNA (referred to as "B3-WT"), shown in Fig. 1a, b. We found up to a 20%

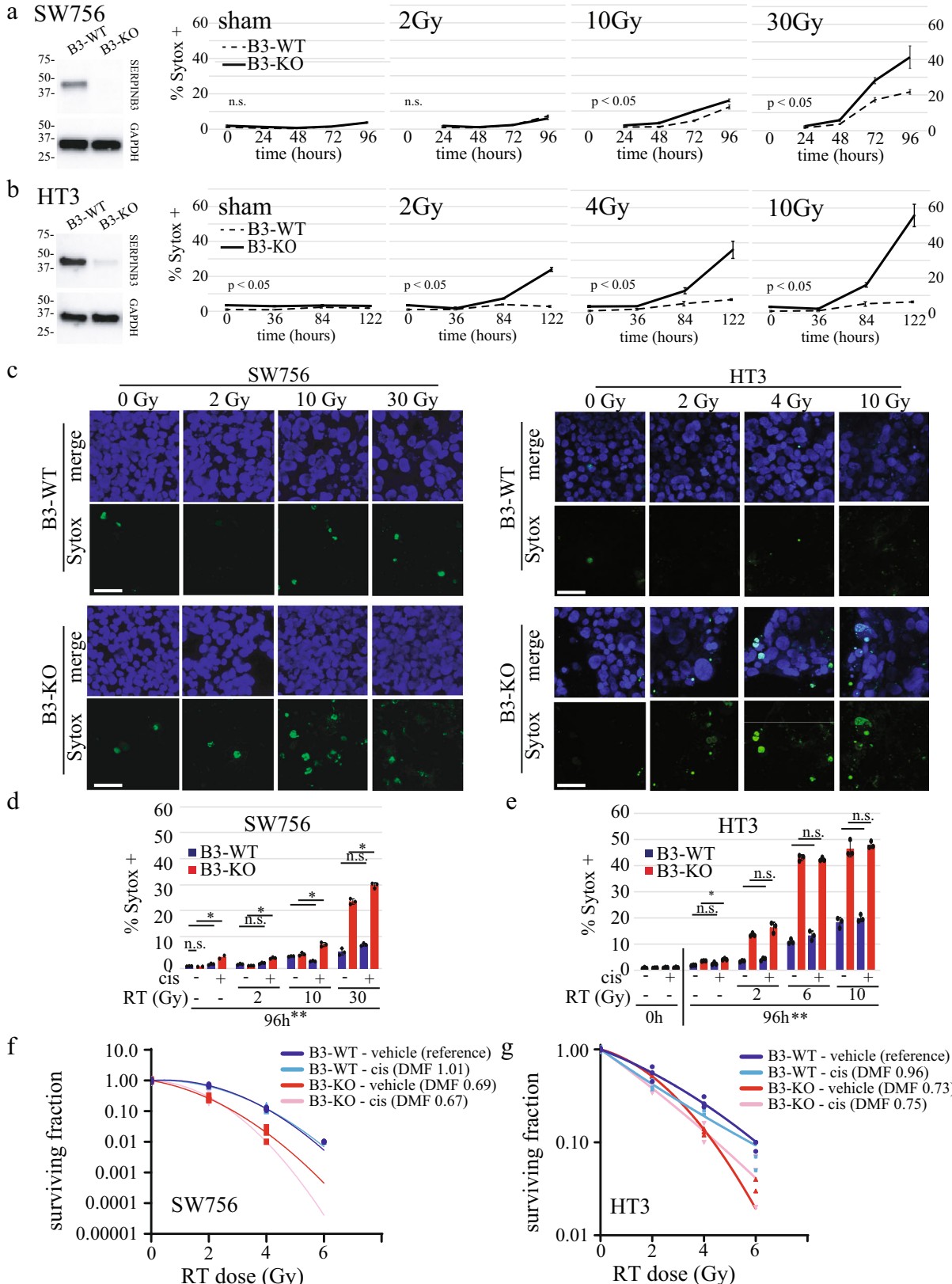

absolute difference in cell death in the SW756 background, and 50% in the HT3 background (Fig. 1a–c). We then compared the effect of single gene knockout of *SERPINB3* on radiosensitivity with or without cisplatin chemotherapy, the radiosensitizing agent used as standard of care for the treatment of patients with cervical cancer. We found that the effect of B3-KO on both cell death (Fig. 1d–e) and clonogenic survival (Fig. 1f–g) was greater than the radiosensitizing effect seen with cisplatin chemotherapy. Dose modifying factors (DMF, determined as the ratio of the dose resulting in 10% surviving fraction compared to control untreated) was greater for single gene *B3*-KO compared to cisplatin in both SW756 (DMF 0.69 for B3-KO versus 1.01 for

**Fig. 1 SERPINB3-KO cells are more sensitive to radiation and cisplatin-induced cell death. a, b** Western blot showing SERPINB3 in B3-WT and B3-KO cell lines. Percent Sytox-positive CRISPR-Control (B3-WT) and SERPINB3-KO (B3-KO) cells at indicated time points after 2, 10, or 30 Gy RT in SW756 (**a**) and 2, 4, or 10 Gy in HT3 (**b**). *P*-value displays sample variation for two-way ANOVA with replicates over time, n.s. not significant (bar = mean of triplicate wells, 2–4 fields of view per well, error bars = standard deviations, one representative of three biologic replicates shown). **c** Representative confocal fluorescent images of SW756- and HT3-B3-WT and -B3-KO Sytox (pseudo-colored green) and merge of Hoescht (pseudo-colored blue) and Sytox images at 84 h post-treatment. Scale bar indicates 50 μm. **d, e** Percent Sytox-positive cells following treatment with increasing doses of RT with or without cisplatin at 96 h in SW756 (**d**) and HT3 (**e**) background (bar = mean of triplicate wells, 2–4 fields of view per well, error bars = standard deviations, one representative of three biologic replicates shown). Student's *t*-test *p*-values shown, n.s. = *p*-value not significant, * = *p* < 0.05, ** = *p* < 0.05 for all comparisons of B3-WT vs B3-KO, except where indicated. **f, g** Clonogenic survival of B3-WT or B3-KO cells treated with increasing doses of RT with or without cisplatin in SW756 (**f**) or HT3 (**g**) background. Individual data points from a representative experiment (one of three biologic replicates) with fit linear quadratic equation. DMF for 10% surviving fraction is shown relative to that of Control cells treated with vehicle (reference).

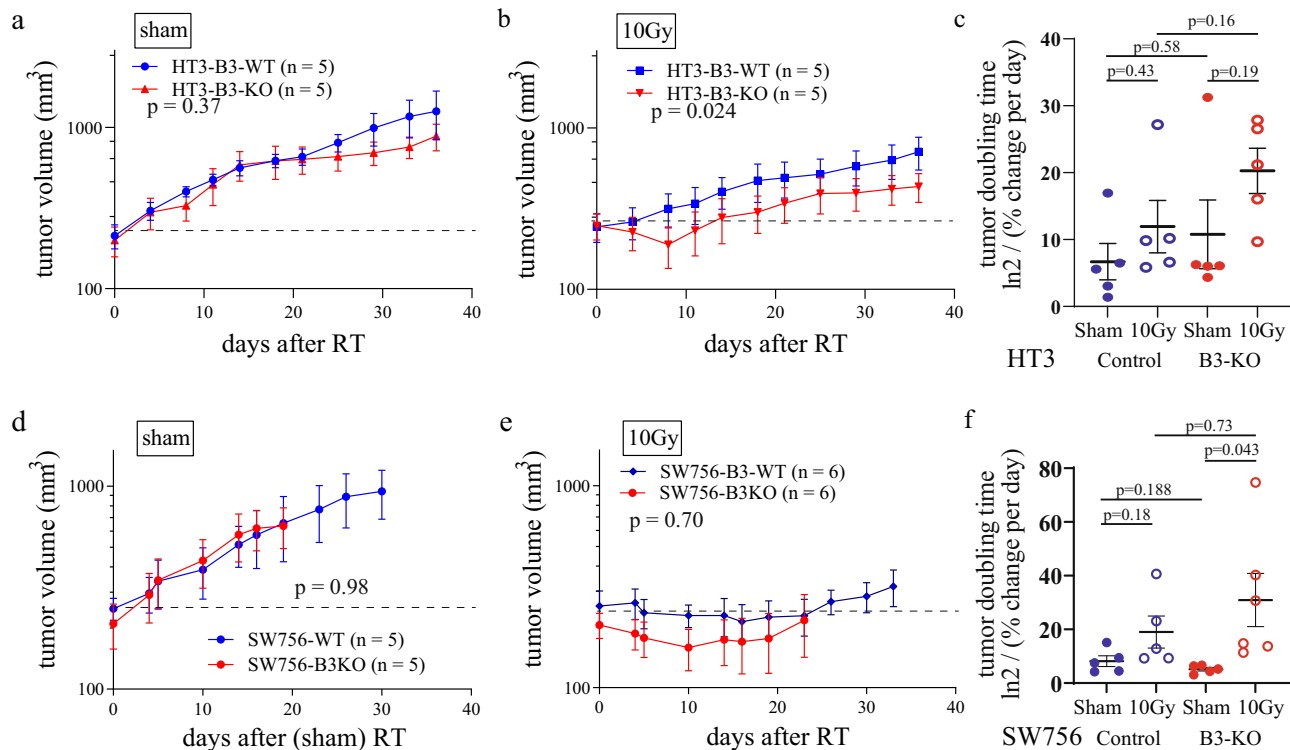

**Fig. 2 SERPINB3-KO tumors are more sensitive to radiation in vivo. a, b** Mean tumor volume with standard error is plotted over time after RT for HT3-B3-WT and –KO flank tumors treated with sham (**a**), or 10 Gy RT (**b**). Number of mice in each group as indicated in the legend, dashed line at mean volume at time of RT. Two-way ANOVA analysis *p*-value displayed on each graph. **c** Calculated tumor doubling (ln2/(% volume change per day)) is shown for HT3-B3-WT and –B3-KO cells after sham or 10 Gy. Individual tumors with mean and standard error are shown. *P*-values from paired *t*-test comparisons are shown. **d, e** Mean tumor volume with standard error is plotted over time after RT for SW756-B3-WT and –KO flank tumors treated with sham (**d**), or 10 Gy RT (**e**). Number of mice in each group as indicated in the legend, dashed line at mean volume at time of RT. Two-way ANOVA analysis *p*-value displayed on each graph. **f** Calculated tumor doubling (ln2/(% volume change per day)) is shown for SW756-B3-WT and –B3-KO cells after sham or 10 Gy. Individual tumors with mean and standard error are shown. *P*-values for paired *t*-test comparisons are shown.

cisplatin) and HT3 (DMF 0.73 for B3-KO versus 0.96 for cisplatin), and cisplatin did not further sensitize B3-KO cells to radiation (DMF 0.69 vehicle versus 0.67 cisplatin for HT3, and 0.73 vehicle versus 0.75 cisplatin for SW756). Cisplatin did reduce plating efficiency of HT3 but not SW756 cells (Supplemental Fig. 1c, d), indicating a cytotoxic effect in the absence of radiosensitization, which has been previously reported with cisplatin in cervix cancer cell lines[23].

**SERPINB3-KO tumors are more sensitive to radiation in vivo**. To determine the role of *SERPINB3* in a p53 mutant cervix tumor in vivo, we established subcutaneous flank xenografts of the HT3 cell line in female athymic nude mice, with either B3-WT or B3-KO clones. Sham-treated HT3-B3-WT and –B3-KO tumors grew at a similar rate in vivo (Fig. 2a). Not surprisingly, HT3-B3-WT

tumors were resistant to radiation, and after a single high dose but clinically relevant fraction of 10 Gy (recapitulating high dose rate brachytherapy) we observed no tumor regression (Fig. 2b). All but 1 of the tumors doubled in size by three weeks after irradiation (Supplemental Fig. 1e). In contrast, all but one HT3-B3-KO xenografts showed tumor regression followed by sustained tumor growth delay (Fig. 2b), and less than 50% of the tumors doubled over the course of the experiment (Supplemental Fig. 1e). Additionally, four of five tumors in the irradiated HT3-B3-WT group also developed ulcers requiring sacrifice compared to none of the irradiated HT3-B3-KO tumors. Calculated tumor doubling time (ln 2/(% volume change per day)) as described by *Lee* et al.[24] for individual xenografts is shown in Fig. 2c.

SW756-B3-KO xenograft tumors took longer to reach the target size for radiation treatment, but grew at a similar rate to

SW756-B3-WT tumors once established (Fig. 2d). SW756 tumors were more radiosensitive; a single fraction of 10 Gy induced tumor growth delay in both B3-WT and B3-KO tumors with B3-KO tumors regressing in size prior to delayed regrowth (Fig. 2e). Fewer radiated B3-KO tumors reached pre-radiation size compared to B3-WT tumors (Supplemental Fig. 1f), and tumor doubling time of B3-KO irradiated tumors was significantly longer compared to sham-treated tumors, while tumor doubling time of irradiated B3-WT tumors was not different from sham-treated B3-WT tumors (Fig. 2f).

**Increased radiation-induced cell death in B3-KO cells is not explained by cell cycle distribution or compromised repair of DNA double strand breaks**. Sensitivity to radiation-induced death can vary depending on the phase of cell cycle at the time of treatment. Cells in S phase at the time of radiation are typically more radioresistant and those in G2/M are more radiosensitive[25]. Therefore, we analyzed cell cycle distribution of unsynchronized sham-treated B3-WT and B3-KO cells and re-sorting 48 h after

4 Gy by quantifying DNA content (Fig. 3a–d). Sham-treated HT3-B3-KO cells had 7% cells in S-phase compared to 6% in B3-WT (Fig. 3b), and sham-treated SW756-B3-KO cells had 40% of cells in S-phase compared to 32% in B3-WT cells (Fig. 3d). Sham-treated SW756-B3-KO cells had 2% of cells in G2/M phase compared to 5% of B3-WT, and G2/M population was not different in sham-treated HT3 cells. In response to radiation, accumulation of cells in the G2/M phase was not different in B3-WT and B3-KO cells with the SW756 cell-line background (HPV positive, p53 wild-type). Cell cycle distribution did not change following RT for either HT3-B3-WT or B3-KO cells (HPV negative, p53 mutant), consistent with aberrant cell cycle checkpoint response commonly seen in p53 mutant cells (Fig. 3a, b).

Reduced capacity to repair double strand DNA breaks induced by ionizing radiation results in decreased clonogenic survival. Thus, in order to determine if DNA-damage repair capacity contributes to radiosensitivity of B3-KO cells, we quantified gamma-H2AX (γH2AX) foci in B3-WT and B3-KO cells treated with sham or 2 Gy RT at 30 min and 24 h post-treatment as a surrogate for repair of DNA damage. As expected, increased

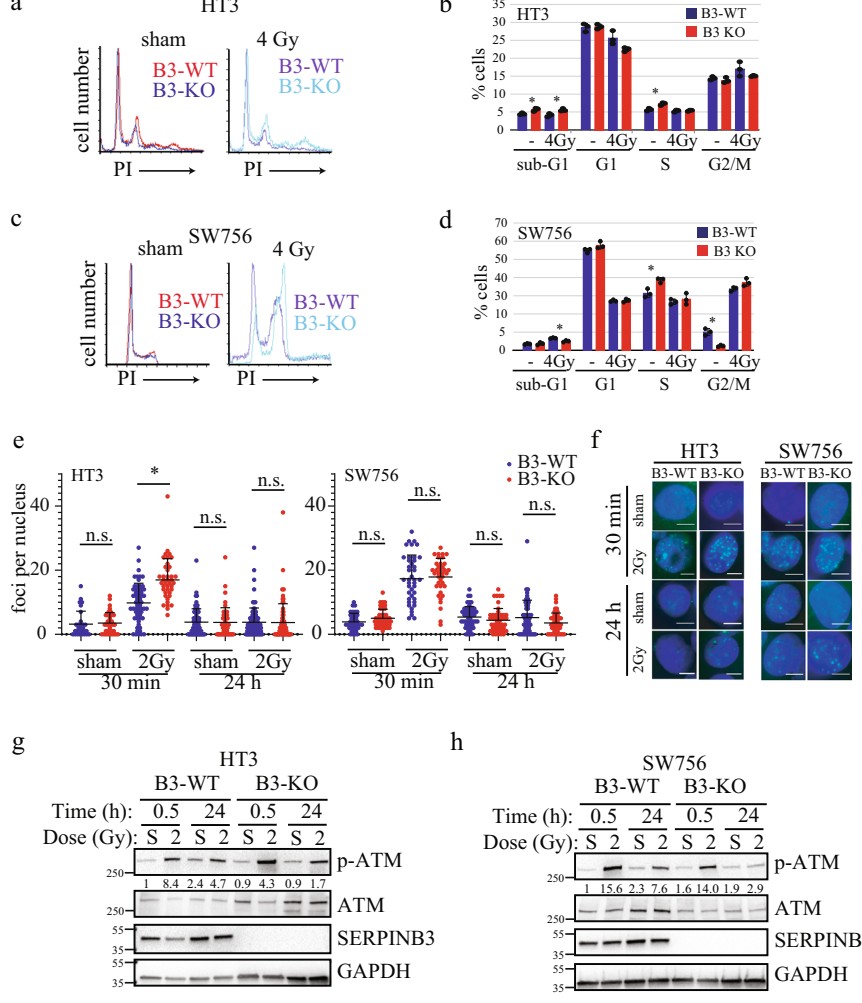

**Fig. 3 Increased B3-KO cell death is not explained by cell cycle distribution or impaired DNA-damage repair. a, c** Representative histograms for DNA content determined by propidium iodide staining and flow cytometry analysis of HT3 (**a**) or SW756 (**c**) cells, with PI intensity on the x-axis. **b, c** Mean and standard deviation quantified cell cycle distribution for triplicate repeats of HT3 (**b**) and SW756 (**d**) experiments. Individual data points are shown, and asterisks indicate students t-test p-value < 0.05. **e** γH2AX foci per nucleus in HT3- and SW756-B3-WT -and -B3-KO cells 30 min or 24 h after treatment with sham or 2 Gy. Dots represent individual nuclei, with mean and S.D. overlay. N.s. not significant, * = p < 0.05. **f** Representative nuclei shown for each condition with Hoechst 33342 (blue) and γH2AX (green) merge. White scale bars shown. **g, h** Western blot analysis of total cell lysates from HT3- and SW756-B3-WT and –B3-KO cells 30 min and 24 h after treatment with sham (S) or 2 Gy radiation. Band intensity of p-ATM normalized to band intensity of GAPDH is displayed under p-ATM blot.

γH2AX foci were observed at the 30 min timepoint in all cell lines (foci/nucleus increased 4–5 fold, Fig. 3e). Foci resolved to sham levels by 24 h in all cell backgrounds, suggesting that the DNA-damage repair machinery is functioning similarly in both B3-WT and B3-KO cells. Interestingly, the mean number of foci per nucleus was moderately but reproducibly higher in HT3-B3-KO cells compared to HT3-B3-WT cells at 30 min post-2Gy, potentially indicating higher direct DNA damage in these cells, or perhaps more likely, different kinetics of response to DNA damage (Fig. 3e). Representative fluorescent images are shown in Fig. 3f. We also evaluated phosphorylation of ataxia-telangiectasia mutated (ATM) by Western blot as an indicator of response to DNA damage; levels of phosphorylated-ATM were increased 30 min after 2 Gy radiation, and decreased by 24 h in all cell backgrounds (Fig. 3g, h). There was no apparent difference in 30 min ATM phosphorylation between HT3-B3-WT and –B3-KO cells.

**Cell death in B3-KO cells following RT suggests potential engagement of multiple cell death mechanisms including apoptosis/caspase-dependent cell death and lysosome-dependent necrosis with features of lysoptosis.** While ionizing radiation is known to induce cell death in tumor cells, the precise mechanism of this demise is unclear and could potentially involve multiple cell death pathways. Thus, we took a multi-faceted approach to determine which cell death mechanism(s) occur in response to radiation in B3-WT and B3-KO cells over time. First, we evaluated cells with transmission electron microscopy (TEM) to determine if cell death morphology induced by ionizing radiation was primarily apoptotic or necrotic. Untreated B3-WT and B3-KO cells showed cells with large nuclear:cytoplasmic ratio as expected for carcinoma cells, without apparent differences in morphology (Fig. 4a, d, g, j). In contrast, when treated with IR, dead B3-KO cells displayed necrotic morphology in both HT3 and SW756 (Fig. 4b, e, h, k), with swollen and disintegrating nuclei (note size of B3-KO 10 Gy nucleus is ~20–30 μM in diameter compared to 10-15μM in sham-treated and B3-WT 10 Gy conditions), and highly vacuolated cytoplasm. High magnification images show breaks in the plasma membrane, cytoplasmic clearing, and largely normal mitochondrial morphology (Fig. 4f, l). LDH release was detected in irradiated B3-KO cells at similar time points, consistent with plasma membrane permeability (Supplemental Fig. 1g). Irradiated HT3-B3-WT and SW756-B3-WT cells displayed enlarged cells with more frequent mitochondria, intact nuclear envelop (Fig 4c, i). Blinded scoring of randomly sampled TEM images by two reviewers shows a higher proportion of cells with necrotic morphology in irradiated B3-KO cells (~25%) compared to irradiated B3-WT cells (<5%) (Fig. 4m). Vacuoles seen in the irradiated B3-KO cells were membrane-bound, consistent with lysosomes, and in many cases these membranes were ruptured, releasing their contents into the cytoplasm (Fig. 4n–q).

Second, cells were treated with increasing doses of RT and analyzed by Western blot and fluorescent microscopy at 24 h intervals up to 96 h. Western blots were probed for markers of multiple regulated cell death pathways including cleavage of end-effector caspase-3 and caspase-7 (apoptosis), and Poly (ADP-ribose) polymerase (PARP) cleavage products (apoptosis), gasdermin D (GSDMD) cleavage to p30 pore-forming product (pyroptosis), and phosphorylation of mixed lineage kinase domain like pseudokinase (MLKL) (necroptosis), and receptor interacting serine/threonine kinase 3 (RIPK3) (necroptosis). Of note, there are no accepted markers of ferroptosis or lysoptosis that can be determined by Western blot. Cell death was quantified in parallel using Sytox exclusion and positive controls for caspase and PARP cleavage (etoposide [85 μM]), GSDMD cleavage (electroporated LPS [12.5 μg/mL]), and MLKL/RIPK3

phosphorylation (TNFα [20 ng/mL] + BV6 [100 nM] + ZVAD [50 μM]) were performed simultaneously (Fig. 5a, b, and Supplemental Fig. 2a). In HT3 cells, caspase-3 (CC-3), caspase-7 (CC-7) and PARP cleavage was detected at later time points with an apparent dose response (Fig. 5a, Supplemental Fig. 2a). There was qualitatively more caspase-3/7 cleavage in HT3-B3-KO cells compared to B3-WT cells, though low compared to the amount of cell death seen at those time points and compared to the apoptosis positive control (Fig. 5a, Supplemental Fig. 2b). In SW756 cells, no caspase-3 or caspase-7 cleavage was detected even 96 h after 30 Gy (Fig. 5b). We found no evidence of GSDMD cleavage, or p-MLKL, p-RIPK3/RIPK1 (markers of pyroptosis and necroptosis, respectively) in either HT3 or SW756 cells (Fig. 5a, b). Additional controls to demonstrate detection of markers by Western blot are shown in Supplemental Fig. 2b-d, including treatment with inducers and inhibitors of apoptosis (Supplemental Fig. 2b), necroptosis (Supplemental Fig. 2c), and pyroptosis (Supplemental Fig. 2d).

As radiation therapy-induced senescence can also contribute to decreased clonogenic survival, we evaluated if markers of senescence were differentially induced by RT in B3-WT and B3-KO cells. Sham-treated SW756-B3-KO cells had slightly higher percent positive senescence associated beta-galactosidase (SA-βgal) cells compared to SW756-B3-WT cells (~15% versus ~10%) 24 h after plating (sham, 0 h, Fig. 5c, d); however, there was no difference 96 h after sham-treatment, and SA-βgal-positive cells were not significantly increased in either SW756-B3-WT or –B3-KO cells after 30 Gy RT (Fig. 5c, d). HT3 cells did not have any SA-βgal-positive cells in any fields of view either with sham or 10 Gy RT (Supplemental Fig. 3). Similarly, Western blot analysis for BCL-2 and BAX, which are commonly increased and decreased, respectively, in senescent cells, were not different between B3-WT and B3-KO cells, and did not change in either direction following treatment with ionizing radiation (Fig. 5e).

Thirdly, pharmacologic inhibitors of pan-caspase activity (qVD-OPH [1uM]), necroptosis (necrostatin-1 [10 μM]), pyroptosis (VX-765, 50 μM), and ferroptosis (ferrostatin-1 [50 nM]) were incubated with cells beginning 24 h after treatment with ionizing radiation. The pan-caspase inhibitor qVD-OPH attenuated cell death in both B3-WT and B3-KO cells by about 10% in HT3 cells, but not SW756 cells (Fig. 5f). Necrostatin-1, ferrostatin-1, and VX-765 did not inhibit radiation-induced cell death in either B3-WT or B3-KO cells in either cell background (Fig. 5f). Higher concentrations of ferrostatin-1, a potent lipid radical-trapping antioxidant, did not inhibit radiation-induced death in either B3-WT or B3-KO cells, despite effective complete inhibition of cell death induced by the ferroptosis-inducer erastin (Supplemental Fig. 4a). Similarly, chelation of intracellular iron using deferoxamine did not inhibit radiation-induced cell death in either B3-WT or B3-KO cells (Supplemental Fig. 4b). Given known cellular requirement of iron for homeostasis and eventual toxicity, the deferoxamine was added at various time points after radiation, with the goal of determining requirement of iron for cell death.

Since caspase-3 and caspase-7 cleavage was detected in vitro, we investigated hallmarks of apoptotic cell death in tumors in vivo following radiation. Flank xenografts were established in nude athymic mice from the HT3 cell lines as described above, randomized to sham or 10 Gy radiation, and harvested 96 h post-RT. Percent TUNEL positive nuclei and IHC for cleaved-caspase-3 (CC-3) was quantified on tissue sections. Absolute levels of TUNEL positive nuclei (Fig. 5G, representative images Fig. 5H-K) and CC-3-positive cells (Fig. 5l, representative images Fig. 5m-p) were and remained low (<~10%) in both B3-WT and B3-KO cells.

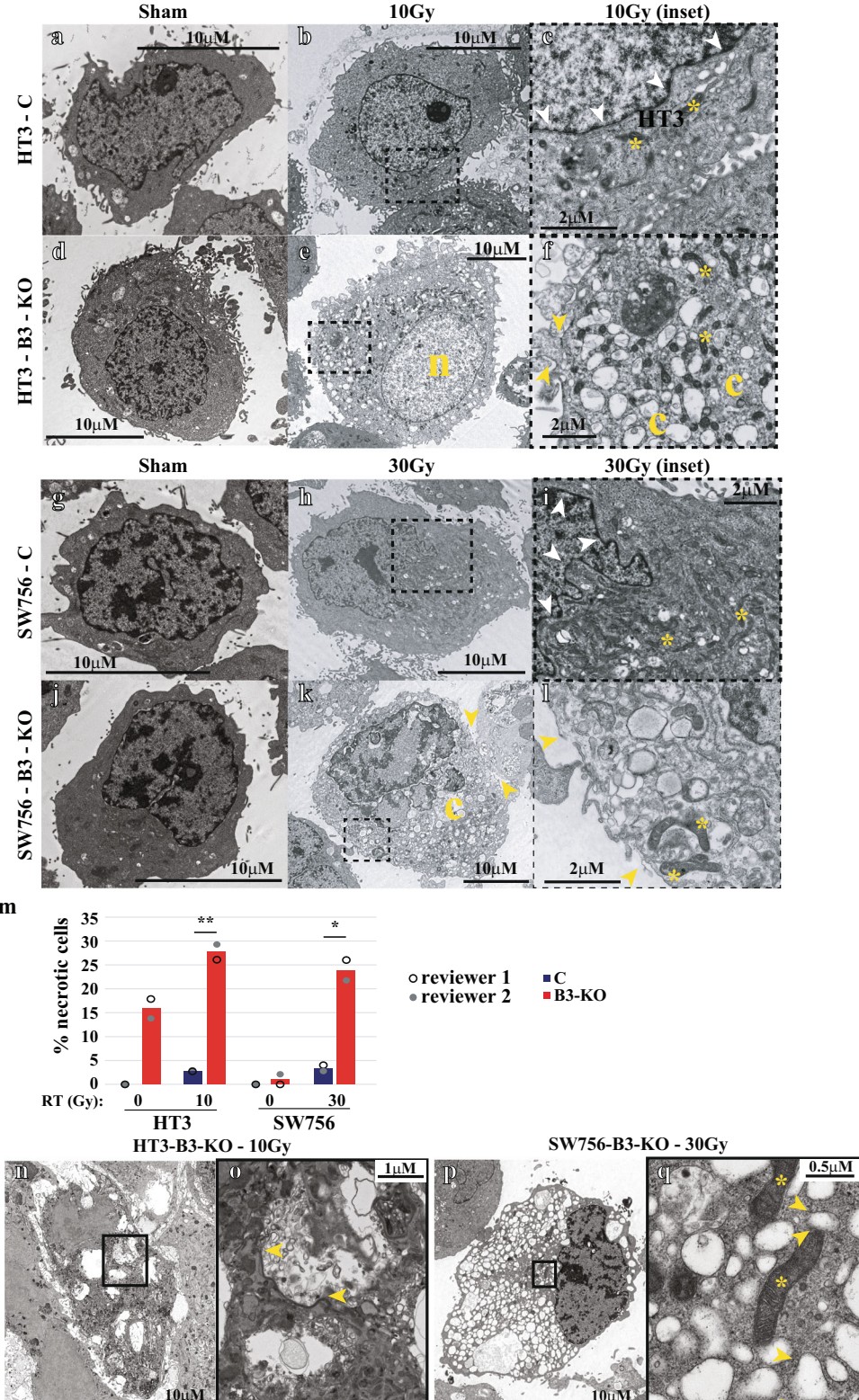

**Fig. 4 Cell death in B3-KO cells following RT is primarily necrotic. a–m** Transmission electron micrographs of HT3 (**a–f**) and SW756 (**g–l**) B3-WT and B3-KO cells 96 h after treatment with Sham or indicated doses of RT. Black scale bars shown for each image. Inset (**c**, **f**, **i**, **l**) shows magnified images of irradiated cells and are bound by a dashed black box. Yellow arrows indicate breaks in the plasma membrane, yellow c is cytoplasmic clearing, yellow asterisks indicate mitochondria, and yellow n indicates devolving nucleus. **m** Quantitation of percent of total cells showing necrotic morphology on 20 random TEM images by two blinded reviewers. Histogram height indicates mean of reviewers, * = student's *t*-test *p*-value < 0.05, ** = student's *t*-test *p*-value < 0.01. **n–q** Representative transmission electron micrographs of HT3-B3-KO (**n**, **o**) and SW756-B3-KO (**p**, **q**) cells 96 h after treatment with RT, with insets (**o**, **q**) showing high magnification regions delineated by a black box. Cytoplasmic lysosome-like structures with ruptured membranes are indicated by yellow arrows. Mitochondria marked with yellow arrows.

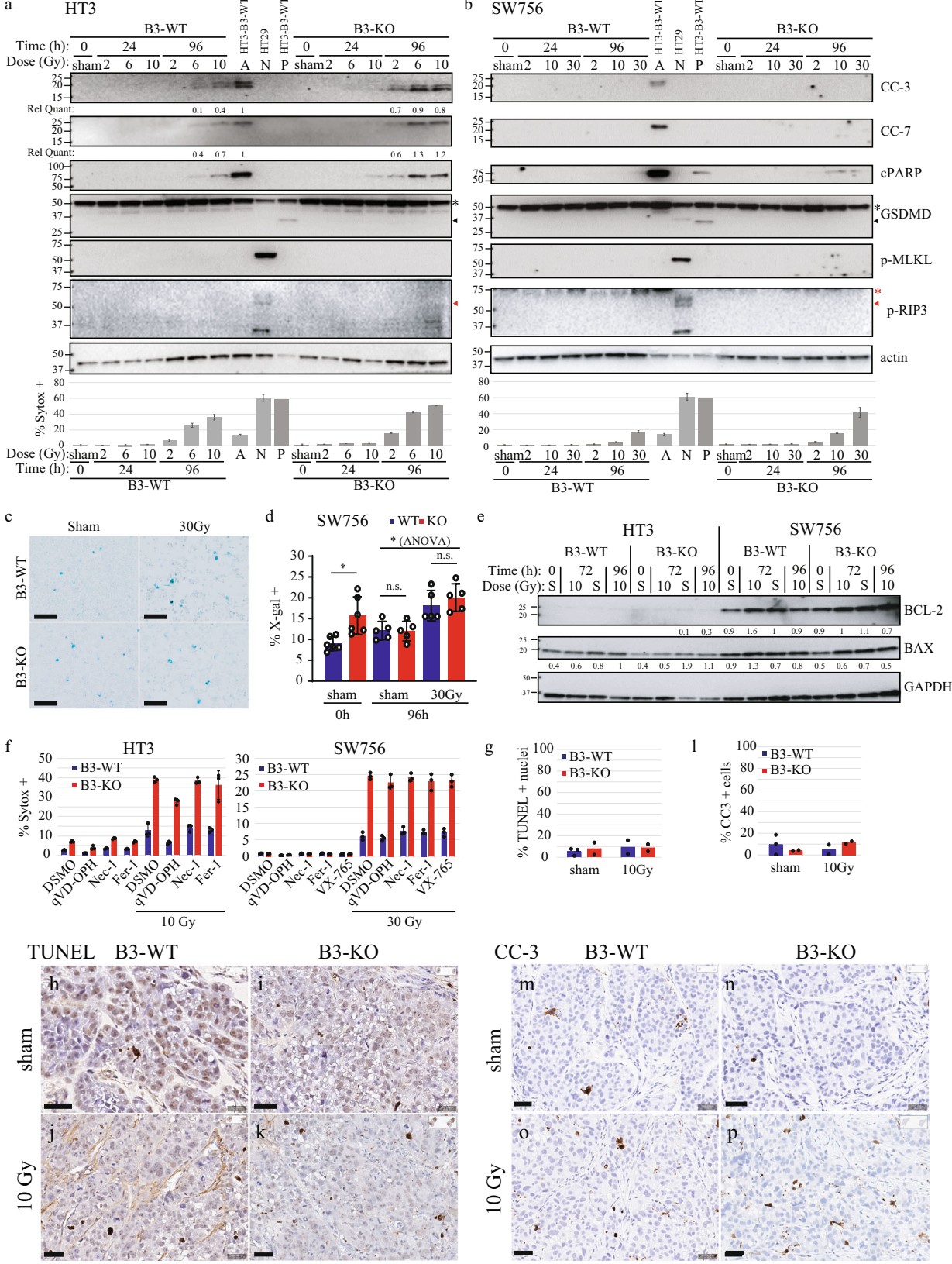

**Radiation induces loss of lysosomal membrane integrity that precedes cell death in SERPINB3-deficient cells.** *C. elegans* genetically lacking *srp-6* (homologue of human SERPINB3) demonstrate extensive lysosomal membrane permeability in their intestinal epithelial cells leading to intestinal cell death and ultimately organismal death when exposed to various cytotoxic stressors[15]. This cell death mechanism has been further characterized and termed lysoptosis (Good et al, "Lysoptosis is an ancient and evolutionarily-conserved cell death pathway moderated by intracellular serpins," manuscript # COMMSBIO-20-2781-T). Given the conserved molecular function of SERPINB3 as an intracellular lysosomal cysteine protease inhibitor, we

**Fig. 5 Cell death in B3-KO cells following RT suggests potential engagement of multiple cell death mechanisms including apoptosis/caspase-dependent cell death and lysosome-dependent necrosis with features of lysoptosis. a** Western blot of total cell lysates from HT3-B3-WT, -B3-KO cells at indicated time points following sham or increasing doses of RT. Positive control lysates from HT3-B3-WT cells treated with etoposide (A for apoptosis), or nucleofected LPS (P for pyroptosis), and HT29 cells treated with TNFα, BV6 and ZVAD-fmk (N for necroptosis) are shown in the center of the blot. Relative intensity of cleaved-caspase-3 (CC-3) and cleaved-caspase-7 (CC-7) is shown normalized to intensity of actin bands, using the positive control for apoptosis as reference. Histogram shows % Sytox-positive cells for each condition determined in parallel (bar = mean of triplicate wells, 2–4 fields of view per well, error bars = standard deviations). Full length GSDMD indicated by black asterisk, and p30 cleavage product indicated by black arrow. Phospho-RIPK3 band (~60 kDa) is indicated by a red arrow, ~75 kDa band on SW756 blot (indicated by red asterisk) is residual PARP antibody signal from previous blotting. **b** Western blot of total cell lysates from SW756-B3-WT, -B3-KO cells at indicated time points following sham or increasing doses of RT. Positive control lysates from HT3-B3-WT cells treated with etoposide (A for apoptosis), or nucleofected LPS (P for pyroptosis), and HT29 cells treated with TNFα, BV6 and ZVAD-fmk (N for necroptosis) are shown in the center of the blot. These positive control lysates were generated concurrently with the experiment. Histogram shows % Sytox-positive cells for each condition determined in parallel. **c** Representative light field images of SA-βgal stained SW756-B3-WT or B3-KO cells 96 h after treatment with sham or 30 Gy, scale bars = 50 μm. **d** Quantitation of percent positive X-gal staining cells with individual data points, mean and standard deviation. Statistical tests are t-test comparisons except ANOVA as noted, * = p < 0.05, n.s. = not significant. **e** WB of total cell lysates from HT3-B3-WT or -B3-KO or SW756-B3-WT or –B3-KO cells treated as indicated. Blots were probed with anti-BCL-2, anti-BAX and GAPDH as a loading control, with quantified band intensity below, normalized to GAPDH. **f** Percent Sytox-positive HT3-B3-WT or –B3-KO cells (left) or SW756-B3-WT or –B3-KO cells (right) treated with indicated pharmacologic inhibitors 24 h after sham or radiation. (bar = mean of triplicate wells, 2–4 fields of view per well, error bars = standard deviations, individual data points shown). **g** Percent TUNEL + nuclei / total nuclei determined on TUNEL stained formalin fixed paraffin embedded sections from indicated tumors (an entire tissue section was analyzed for n = 2 mice per group, n = 3 for B3-WT sham). Bar height indicates mean. **h–k** representative images of TUNEL staining. Scale bar = 40 μm. **l** Histogram of percent cleaved-caspase-3 (CC-3) + nuclei / total nuclei determined on IHC with CC-3 antibody stained from indicated tumors (an entire tissue section was analyzed for n = 2 mice per group, n = 3 for B3-WT sham). Bar height indicates mean, * = p < 0.05, n.s. = not significant. **m–p** representative images of CC-3 staining. Scale bar = 40 μm. Full slide digital images are available at: https://app.histowiz.com/shared_orders/2ccb0ce8-e4e1-41d1-8f75-a0e621ced36e/slides/.

hypothesized that cervical tumor cells lacking SERPINB3 would be susceptible to lysosomal membrane permeability and lysosomal rupture following an insult such as ionizing radiation. TEM of irradiated B3-KO cells showed evidence of vesicular membrane rupture reminiscent of lysosomal membrane permeability in both HT3 and SW756 cells (Fig. 4n–q). To determine if lysosomal membrane integrity was lost prior to cell death or occurred as a postmortem event, we performed live-cell time-lapse confocal microscopy to observe cells in the process of dying. Lysosomes were marked with the acidophilic dye LysoTracker™-deep red (pseudo-colored green), and cell membrane permeability as a measure of cell death was determined by propidium iodide (pseudo-colored red). B3-WT or B3-KO cells were treated with 10 Gy and imaged during a period of expected high percent death, beginning ~72 or 96 h after treatment. Dying B3-KO cells treated with radiation lost lysosomal integrity prior to loss of cell membrane integrity (Fig. 6a, c). HT3-B3-WT cells, in contrast, showed a delayed loss of some LysoTracker staining after becoming PI positive (Fig. 6b, d).

Treatment of cells with the cysteine protease inhibitor E64d inhibited radiation-induced cell death in HT3-B3-WT and to a greater degree in HT3-B3-KO cells (Fig. 6e), providing further supporting evidence that lysosomal protease activity is important for cell death, particularly in the absence of SERPINB3.

**The reactive site loop of SERPINB3 and inhibition of cathepsin L activity is required to protect cervical cancer cells against radiation-induced death and to promote tumor growth.** To determine if SERPINB3 is sufficient to increase radioresistance in cervical cancer cells, SiHa (HPV16 + /p53 wild-type) and C33A (HPV-/p53 mutant) cell lines with no detectable SERPINB3 protein were used to generate stable clones expressing the wild-type SERPINB3 (B3) or an empty vector control (VC), with bicistronic expression of green fluorescent protein (Fig. 7a). Clonogenic survival was significantly higher in SiHa-B3 cells compared to SiHa-VC cells (DMF 1.25, Fig. 7b). C33A cells grow in a manner that is partially adherent with a population of viable cells that easily detach from the tissue culture dishes used for clonogenic survival assays, leading to high variability in the assay. Use of soft agar suspension resulted in disaggregated colonies with

similar challenges. Therefore, the CellTiter-Glo® ATP-dependent reagent was employed to estimate cell viability in C33A cells. Indeed, we find that C33A-B3 cells have higher cell viability following radiation treatment compared to C33A-VC cells, with similar findings in the SiHa background using this assay (Fig. 7c).

To determine if the protease-inhibitory function of SERPINB3 is required for radioprotection in these cells, we generated isogenic stable clones expressing SERPINB3 with a single alanine to arginine amino acid substitution at the P14 residue (Schechter and Berger numbering scheme[26]) of the C-terminal RSL, corresponding to amino acid 341 in the hinge region (termed B3-A341R). The resultant protein is a well-characterized mutant that blocks loop insertion after cleavage by target proteases and thus does not inhibit protease activity[7]. For functional analyses, a clone with similar levels of B3-A341R protein expression compared to the B3-expressing clone was selected (Fig. 7a), and clonogenic survival and cell viability was compared to B3- and VC-containing cell lines (Fig. 7b, c). Expression of B3-A341R did not protect cells from radiation and survival was similar to VC-expressing cells (DMF 0.93 compared to VC). Flank xenograft tumors of the C33A cell lines were established to determine effect of B3 and B3-A341R on tumor growth and in vivo radiation resistance. Sham-treated C33A-B3 tumors overall grew faster than VC or B3-A341R tumors (Fig. 7d, spider plots shown in Supplemental Fig. 5). After tumor establishment, half of the mice were randomized to tumor-directed radiation, and B3 tumors continued to grow after a single dose of 10 Gy compared to VC and B3-A341R tumors (Fig. 7e, spider plots shown in Supplemental Fig. 5). Tumor doubling time was not significantly different in sham-treated versus 10 Gy irradiated B3-expressing tumors, whereas tumor doubling time was longer in irradiated VC- and A341R-containing tumors (Fig. 7f). Images of dissected tumors to portray relative tumor size at the time of sacrifice for sham irradiated and 10 Gy treated tumors are shown in Fig. 7g, and spider plots of individual tumor growth is shown in Supplemental Fig. 4. Taken together, these data suggest that SERPINB3 protects cells from ionizing radiation by inhibiting target lysosomal cysteine proteases.

In order to determine the target protease(s) of SERPINB3 in cervix cells, we first evaluated gene expression levels of the

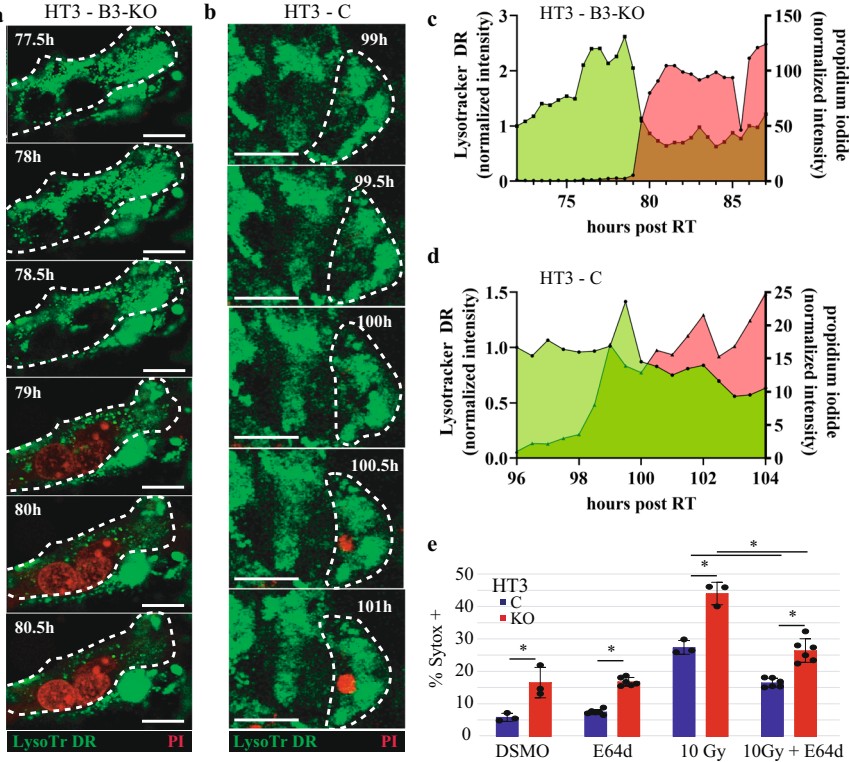

**Fig. 6 Radiation induces loss of lysosomal membrane integrity that precedes cell death. a, b** Live-cell time-lapse confocal pseudocolor images of HT3-B3-KO (**a**) and HT3-B3-WT (**b**) cells at the indicated times after treatment with 10 Gy RT. LysoTracker deep red is pseudo-colored green and propidium iodide is pseudo-colored red. Scale bar = 25 μm. **c, d** cells shown were manually tracked over the timeframe indicated on the x-axis and intensity of LysoTracker deep red (LysoTr) and propidium iodide normalized to cell volume is plotted on the double y-axis. **e** Percent Sytox-positive HT3-B3-WT and –B3-KO cells treated with vehicle DMSO or E64d 24 h after sham or 10 Gy radiation. * = $p < 0.05$ by double-sided t-test. Each data point represents the % Sytox-positive quantified in 2–4 fields of view of one triplicate well ($n = 3$ per condition), and is a representative one of three biologic replicate experiments. Mean indicated by bar height and error bars show standard deviation.

nine mammalian cysteine cathepsin proteases, B cathepsin H, cathepsin C, cathepsin X, also known as cathepsin Z, cathepsin L (CTSL), cathepsin S, cathepsin V, cathepsin F, and cathepsin K. RNA-seq analysis of 68 primary tumor samples collected prior to the start of definitive radiation showed variable expression of cathepsins B, H, C, X, and L (Fig. 7h). Cathepsins B and H are primarily exopeptidases at acidic pH, and cathepsins X and C are exclusive exopeptidases and therefore unlikely to be inhibited by the pseudo-substrate bait and trap mechanism of SERPINB3 (requires endopeptidase activity). Of the five remaining cysteine endopeptidases, we found very low or undetectable levels of cathepsin S, cathepsin V, cathepsin F, and cathepsin K. Western blot analysis of SW756 cell lines showed no detectable cathepsin S and K protein, while CTSL was present (Fig. 7i, positive control for antibodies shown in Supplemental Fig. 6). Therefore, we hypothesized that CTSL was the most-likely target of SERPINB3 in cervix tumor cells and generated stable knockout cell lines using CRISPR-Cas9 targeting of *CTSL* in the SW756-B3-WT and –B3-KO backgrounds (Fig. 7i). SW756-B3-KO/CTSL-KO cells were protected from radiation-induced cell death compared to SW756-B3-KO/CTSL-WT cells, while the absence of CTSL in the SW756-B3-WT background did not impact response to radiation (Fig. 7j).

## Discussion

Since its isolation from human cervical tumors, SCCA has been shown to serve as a strong and consistent indicator of poor outcomes in cervical cancer and other types of cancers. While others have shown that SCCA1/SERPINB3 promotes tumor growth and resistance to cytotoxic agents, these data show that SERPINB3 is directly responsible for radiation resistance of cervical cancer cells by inhibiting a lysosome-mediated necrotic cell death pathway. Moreover, we show that the effect of SERPINB3 loss on radiation sensitivity in cervical tumor cells is similar if not greater than cisplatin, currently used as the standard of care to radiosensitize cervical cancer. These findings suggest that SERPINB3 may serve as a therapeutic target for radiosensitization of resistant cervical cancers. Additionally, the data also implicate a lysosome-dependent regulated cell death pathway termed lysoptosis in radiation-induced cell death and resistance.

In contrast to other lysosome-dependent cell death mechanisms triggered by harsh mechanical disruption of the lysosomal membrane[27], lysoptosis is blocked by protease-inhibitory activity of SERPINB3 in response to physiologic and pathologic stimuli including hypotonic stress, and as we show here, by IR. Importantly, in the companion paper by Luke et al, we show that loss of the cytoprotectant SERPINB3 sensitized cells to lysoptosis-like death in response to diverse cytotoxic agents, including staurosporine, which classically induces apoptosis, $H_2O_2$, which induces MTP-dependent necrosis in SERPINB3-WT cells, and combination treatment (TNFα, SMAC mimetic, pan-caspase inhibitor, and cyclohexamide) expected to induce necroptosis. This suggests that upstream triggers of lysoptosis-like death may converge on unabated lysosomal membrane permeability and activity of lysosomal enzymes including cathepsin L when SERPINB3 is deleted.

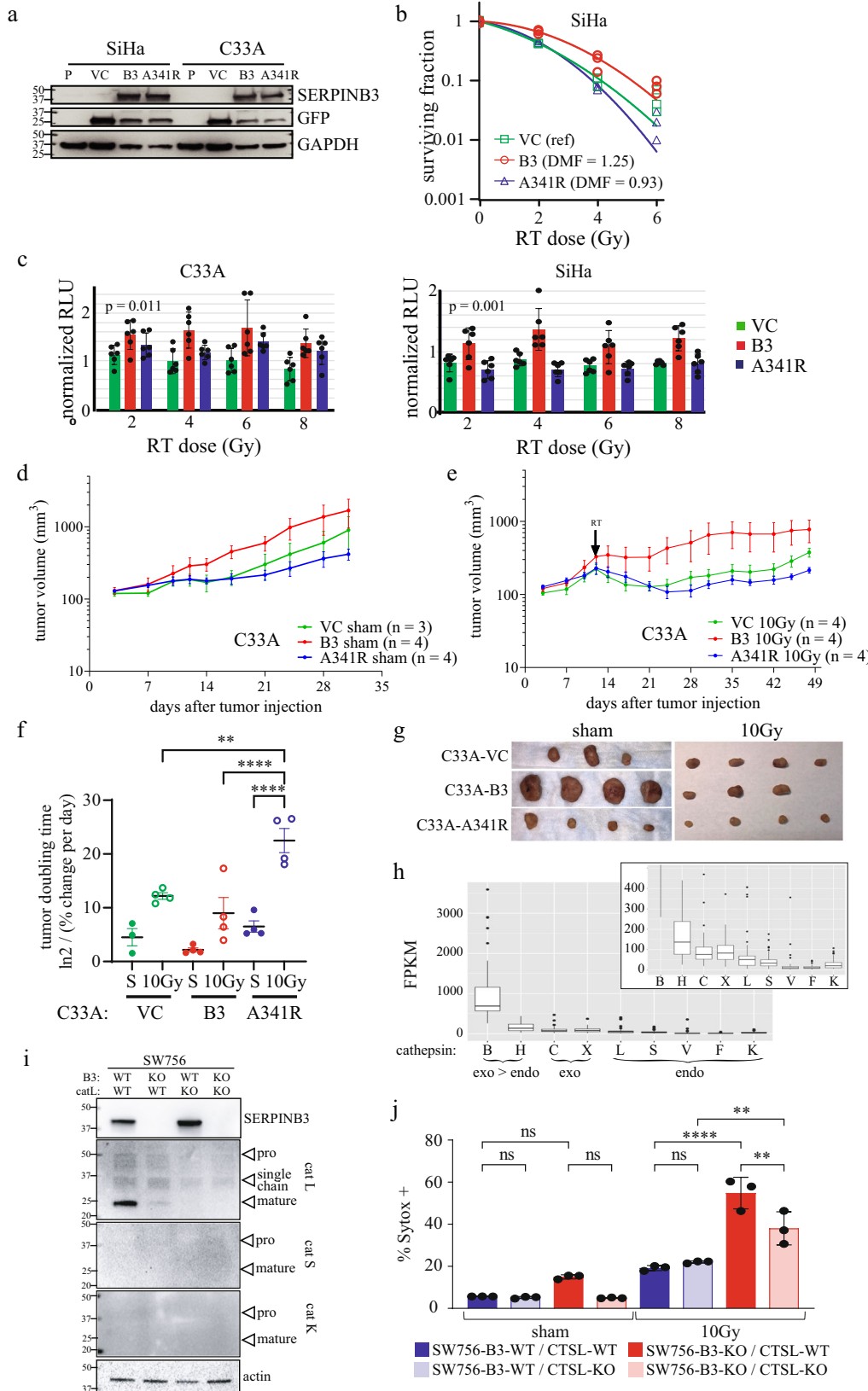

Ionizing radiation has a host of direct and indirect cellular effects on tumor cells ultimately leading to recovery and survival, or cell death. Direct damage to DNA and other macromolecules is known to lead to the induction of apoptosis primarily in hematologic cells, but apoptosis is not a predominant mode of cell death in most solid tumors. Following IR, the inability of the cell to undergo mitosis secondary to dysfunctional cell cycle checkpoints and accumulated unrepaired DNA damage, entering a state known as mitotic catastrophe, thought to ultimately lead to cellular demise by secondary death pathways, or to enter a senescent state[28]. However, the dynamics of radiation-induced cell death, and the exact molecular mechanisms contributing to

**Fig. 7 SERPINB3 is sufficient to protect cervical cancer cells against radiation-induced death, but requires a functional reactive site loop. a** Western blot of SiHa and C33A parental cervical tumor cell lines (P) which do not express SERPINB3 protein at baseline, and stable clones engineered to express wild-type SERPINB3 (B3), SERPINB3-A341R mutant (A341R), or empty vector control (VC). **b** Clonogenic survival of SiHa cells expressing VC, B3 or B3-A341R constructs, following increasing doses of RT, individual data points with fit linear quadratic curve is shown. Dose modifying factor (DMF) displayed with reference to VC. **c** Relative proliferation of C33A cells expressing VC, B3 or B3-A341R treated with RT normalized to Sham-treated cells at 24 h post-treatment, and measured by relative light units (RLU) of CellTiter-Glo luminescent reagent measuring cellular ATP. Individual data points ($n = 6$) are shown mean indicated by bar height and standard deviations are plotted. *P*-value for one-way ANOVA shown. **d, e** Mean tumor volume curves of flank xenograft generated from C33A cells expressing VC, B3-wt or B3-A341R treated with sham (**d**) or 10 Gy tumor-directed radiation (**e**) with tumors harvested at indicated time post-treatment. **f** Time to tumor doubling in days for each of the tumors. Individual data points are shown, tumors not doubling by the termination of the experiment were included as a data point on the last day of the experiment. *P*-value for paired *t*-test between sham and 10 Gy groups for each tumor type are shown. **g** Harvested tumors from sham (left) or 10 Gy (right) irradiated tumors from experiment shown above. **h** Fragments per kilobase of transcript per million mapped reads (FPKM) of cysteine cathepsin protease gene transcripts in 68 patient specimens. CTSB and CTSH have predominantly exopeptidase activity (exo>endo), CTSC and CTSX have exclusively exopeptidase activity (exo), and the remainder have exclusively endopeptidase activity (endo). Inset shows the same data zoomed to 0–500 FPKM. **i** Western blot of SW756 cells with wild-type or knock SERPINB3 (B3-WT and B3-KO, respectively) and wild-type or knockout cathepsin L (CTSL-WT and CSTL-KO respectively) Arrows indicate pro-, single chain, and mature forms of cathepsin enzymes. **j** Representative histogram of percent Sytox + cells 96 h post-treatment with sham or 30 Gy in SW756-B3-WT/KO–CSTL-WT/KO backgrounds. Multiple comparisons from one-way ANOVA is shown for indicated comparisons (ns non-significant, ** = p-value < 0.01, **** = p-value < 0.0001). Individual data points with bar height at the mean % Sytox-positive quantified in 2–4 fields of view of triplicate well ($n = 3$ per condition), and is a representative one of three biologic replicate experiments.

the cell fate decision are complex. The most recent recommendations from the Nomenclature Committee on Cell death describes myriad cell death subroutines, including cell-intrinsic and –extrinsic modes[27]. Importantly, the committee stresses that there is often a great deal of interconnectivity between signal transduction cascades leading to one cell death mode versus another. Therefore, in order to truly understand the relative contributions of various lethal regulated cell death programs to radiation-induced tumor cell kill, it is critical to evaluate not only the presence of individual markers of cell death pathways at single time points and under single conditions, but multiple markers of suspected death modes at varying time points and dose levels. As such, in this study we undertook a broad evaluation to determine the intrinsic cell death routines contributing to radiation-induced death in B3-KO cells compared to B3-WT cells. The goal was to identify vulnerabilities exposed by the loss of SERPINB3 that can be exploited for therapeutic radiosensitization.

The first finding was the prevalence of necrotic morphology in B3-KO cells following ionizing radiation. Although some end-effector caspase cleavage was detected by Western blot analysis, this was only in HT3-B3-C and –B3-KO cells, and not in the SW756 background. Moreover, morphology as determined by the gold standard TEM showed a preponderance of necrosis and very little evidence of apoptotic morphology. Thus, while caspase-dependent cell death is induced by IR in HT3 cells, SERPINB3-knockout exposes a vulnerability to predominantly necrotic death. Pyroptosis is a form of necrotic regulated cell death typically induced by microbial pathogens, whereby cleavage of gasdermin D by caspase-1, or in some cases gasdermin E cleavage by caspase-3, mediates its localization to the cell membrane. Gasdermin D-N or gasdermin E-N oligomerization forms pores resulting in rapid permeabilization of the cell membrane[29]. Despite some caspase-3 cleavage, we found no evidence of gasdermin D or E cleavage following radiation in either B3-WT or B3-KO cells. Ferroptosis is another necrotic regulated cell death mechanism which occurs independently of caspase activation and relies heavily on generation of reactive oxygen species, a hallmark of ionizing radiation[30]. Nevertheless, ferrostatin-1 did not inhibit radiation-induced cell death in either B3-WT or B3-KO cells, and TEM images of necrotic cells revealed largely intact mitochondria suggesting ferroptosis is not a predominant form of radiation-induced cell death in B3-KO cells.

In *C. elegans*, the homolog of SERPINB3, srp-6, serves as an intracellular cysteine protease inhibitor and was previously shown to protect animals against diverse toxic stressors by inhibiting lysosomal proteases and subsequent animal death[15]. In addition to hypo-osmotic conditions and heat shock, we also demonstrated that *srp-6* null animals were more sensitive to oxidative stress and hypoxia. Additionally, several groups have shown that DNA damage induced by topoisomerase inhibitors camptothecin and etoposide causes lysosomal membrane permeability[31,32]. Therefore, we hypothesized that radiation, which causes both DNA damage and accumulation of oxidative species, might induce lysosomal membrane permeability in cervical cancer cells, particularly in cells lacking SERPINB3. Indeed, we observed TEM evidence of lysosome-like vesicle membrane rupture in B3-KO tumor cells treated with ionizing radiation, and loss of lysosomotropic staining by live-cell microscopy in dying cells, similar to that seen in *srp-6* null *C. elegans*. Further supporting the idea that SERPINB3 lysosomal protease inhibitor activity is responsible for its ability to protect cells against radiation, we found that the pharmacologic cysteine protease inhibitor E64d abrogated cell death in B3-KO cells. Additionally, expression of wild-type but not the RSL-mutant B3-A341R protected SERPINB3-low cervical cancer cells from radiation further supporting the importance of an intact RSL to bait-and-trap target lysosomal proteases released from the lysosome after exposure to ionizing radiation. Guided by expression patterns in primary cervix tumor samples, we identified cathepsin L as at least one of the cysteine protease targets of SERPINB3 in cervical tumor cells that mediates radiation-induced cell death in the absence of SERPINB3.

While the DNA-damage repair capacity appears intact in B3-KO cells, suggesting that inability to repair DNA damage is not contributing to the radiosensitivity observed in these cells, we did observe that the initial quantity of γH2AX foci, which is a surrogate for number of double strand DNA breaks, was higher in B3-KO cells specifically in the HT3 background. The mechanism of this is not clear and is under current scrutiny; however, the myeloid and erythroid nuclear termination protein, a chicken Clade B or intracellular SERPIN, is associated with heterochromatin[33], thus there is precedence for nuclear function of SERPIN intracellular family proteins. Moreover, cysteine cathepsin proteases have been reported to localize to the nucleus[34,35]. It is conceivable that nuclear cathepsins may have a role in vulnerability to DNA damage that is modifiable by the presence or absence of SERPINB3. Perhaps more likely is a slightly different kinetics of foci formation in the B3-WT and B3-KO cells that will be discerned in future detailed analysis. Similarly, we observed

some differences in baseline cell cycle distribution between B3-KO and B3-WT cells. While these differences do not appear to explain increased radiation sensitivity of B3-KO cells, since they would predict B3-KO cells would be more radioresistant, this observation is a focus of ongoing studies.

Importantly, we found similar effects of SERPINB3 on radiation-induced cell death and resistance in cervical tumor cell lines that are HPV positive and negative, and p53 wild-type and mutant. These findings suggest that radiation-induced lysosome-mediated necrosis proceeds in a manner that is not dependent on functional p53.

The major limitation of the current study is the experimental difficulty posed by the long time course of radiation-induced death. Instead of minutes to hours required for classic cell death inducers to result in cellular demise, most non-hematologic cells treated with IR do not die for at least 48 h after treatment, and in many cases several days after a single treatment. While we made every effort to capture an accurate view of cell death modes during each phase of this long time course, and in response to various doses of radiation, it is possible that less prevalent cell death events were not captured by Western blot, live-cell imaging and TEM assays. Additionally, the time required to observe a single cell death event using live-cell imaging inherently introduced difficulty in tracking cells while maintaining confocal resolution. Finally, the complexity of signal transduction occurring in the hours to days after treatment with ionizing radiation and the long time course of the experiments likely complicates the interpretation of pharmacologic inhibitors of cell death.

While we are currently investigating the possibility, it remains unclear if other cancer subtypes expressing SERPINB3 are protected against radiation in a similar manner. Given the available evidence that SERPINB3 expression in many other cancers including epithelial cancers of the head and neck, anus, lung, esophagus as well as hepatocellular carcinoma carries a similar poor prognosis, we expect this to be the case.

Taken together, this body of evidence contributes to the understanding of regulated cell death occurring in tumor cells in response to therapeutic ionizing radiation, and identified a targetable vulnerability, SERPINB3.

## Methods

**Guidelines and regulatory approvals.** The methods were performed in accordance with relevant guidelines and regulations and approved by the Washington University Institutional Biological & Chemical Safety Committee under protocol #12737 (Ver.2.1). All mouse experiments were performed in accordance with relevant guidelines and regulations and approved by the Washington University Institute Institutional Animal Care and Use Committee (Protocol #20-0470). Human subjects research was approved with informed consent by the Washington University Institutional Review Board (No. 201105374).

**Cell lines and tissue culture.** All cell lines were obtained from ATCC and grown in monolayer at 37 °C/5% CO$_2$ in Dulbecco's modified Eagle's medium (DMEM) or Iscove's Modified Dulbecco's Medium (IMDM) supplemented with 10% fetal bovine serum, 0.1 mg/mL penicillin, 100 units/mL streptomycin, and 15 mM Hepes. Generation and characterization of B3-KO and isogenic CRISPR-control cell lines (B3-WT) SW756 and HT3 was described previously[9]. In brief, a single-vector lentiviral system was used, driving expression of Cas9 and the gRNA sequence. KO was confirmed by sequencing of the *SERPINB3* gene, Western blot showing no protein product, and sequencing of most-likely off-target genes to confirm specificity. SW756-B3-WT/cathepsin L knockout (B3-WT/CTSL-KO) and SW756-B3-KO/cathepsin L knockout (B3-KO/CTSL-KO) lines were derived from SW576 parental and −B3-KO clonal line, respectively, using a CTSL-targeting guide RNA (GTGAGGAATCCTATCCATATG) in exon 5. Knockout was confirmed in the pool cell line by Western blot and next generation sequencing, verifying *CTSL* indels at a high percentage, resulting in a partial knockout cell line. For the current paper, SiHa and C33A cell lines were engineered to stably express pULTRA (Addgene 24129) mammalian vector driving expression of the wild-type *SERPINB3* gene, or *SERPINB3* mutant encoding a single alanine to arginine substitution at amino acid 341 (B3-A341R).

**Irradiation of cell lines and mice.** Cells in dishes were irradiated at a dose rate of ~300 cGy per minute using the RS-2000 Biological System (Rad Source, Suwanee, GA), calibrated once monthly by a medical physicist. Sham irradiation conditions were transported to the irradiator and placed at room temperature conditions while delivering radiation. Mouse xenograft tumors were irradiated using the Xstrahl Small Animal Radiation Research Platform (SARRP) 200 (Xstrahl Life Sciences, Suwannee, GA). After being fitted with a nose cone, mice were individually subjected to isoflurane anesthesia and imaged by on-board micro-computed tomography (CT). CT images imported into Muriplan were used to select an isocenter. The tumor was then irradiated using anterior-posterior opposed beams using the 10 × 10 mm collimator at a dose rate of 3.9 Gy/min. Sham irradiated mice were transported to the SARPP facility and remained in the room at the same conditions during radiation of the irradiated mice.

**Cell cycle analysis.** Cells were seeded at $0.15 \times 10^6$ cells per well and treated 24 h later with 4 Gy or Sham RT. Cells were fixed and permeabilized with Triton-X-100 48 h later (after optimization of IR dose and incubation time), stained with propidium iodide, and cell cycle distribution determined using flow cytometry of DNA content. FlowJo™ Software (Becton, Dickinson and Company, Ashland, OR, USA) was used to analyze and visualize the data.

**Gamma-H2AX foci formation assay.** Cells were seeded in 4-well chamber slides (Nunc® Lab-Tek® II Chamber Slide™, C6807, Millipore Sigma, St Louis, MO, USA) to 50-60% confluence and allowed to adhere to the slide for 24 h. Cells were then treated with sham or 2 Gy radiation as described in the Irradiation of cell lines and mice section and incubated at 37 °C for 30 min or 24 h prior to washing with PBS and fixation with 3% PFA. Cells were then permeabilized with 0.1% Triton-X and washed with PBS before blocking with blocking buffer (10% FBS, 0.5% BSA in PBS) for 1 h at room temperature. Slides were incubated at 4 °C overnight with 1:300 anti-gH2AX antibody (05-636, Millipore Sigma, St Louis, MO, USA). Slides were washed in three changes of PBS and incubated with 1:500 anti-mouse-IgG AlexaFluor 488 conjugated secondary antibody for 3 h at room temperature. Slides were mounted with VECTASHIELD® HardSet™ Antifade Mounting Medium with DAPI (H-1500, Vector Laboratories, Burlingame, CA, USA). Fluorescent images were obtained on the Zeiss LSM510. Images were then analyzed using ImageJ Software to quantitate foci per nucleus for a random selection of six 100X fields of view, corresponding to 30–50 cells per condition.

**Cell death assay.** Cells were seeded in order to achieve 50–70% confluence 24 h later in a 96-well glass-bottom, opaque-walled dish (Greiner Bio-One™ CellStar™ µClear™ 96-Well, Cell Culture-Treated, Flat-Bottom Microplate, Fisher, 7000166). 24 h after plating, plates were treated with varying doses of IR or Sham, with 1 h pre-incubation of inhibitors where indicated. Cisplatin was added 1 h prior to radiation, where indicated, at the half-maximal inhibitory concentration (IC50) of cisplatin monotherapy determined by Alamar blue assay (0.5µM for SW756 and 0.01µM for HT3). At established time points, Sytox™ orange (1:30,000, Fisher Scientific, S11368) and Hoescht 33342 (1:2000) were added 15–30 min prior to quantitative imaging on the Cytation™ 5 multi-mode reader (BioTek® Instruments, Inc., Winooski, Vermont). Percent dead cells was determined by optimized automated counting of Sytox-positive cells divided by Hoescht-positive nuclei, averaged over triplicate wells.

**Western blot.** Cells were seeded and treated as in the cell death assay methods, except scaled to six-well or 10 cm dishes in order to obtain adequate cell lysate. At established time points, coinciding with the cell death assay, cells were lysed with Cell Lysis Buffer (Cell Signaling Technology, Danvers, MA) supplemented with proteinase/phosphatase inhibitors and PMSF. Equal parts protein (25–30 µg) and Laemmli sample buffer (Santa Cruz Biotechnology, Dallas TX) were boiled at 95 °C for 10 min and gel electrophoresed on 4–20% gradient gels (Mini-Protean TGX, Bio-Rad, Hercules, CA), transferred to PVDF blot using the Trans-Blot Turbo Transfer system (Bio-Rad, Hercules, CA), blocked with 5% milk:TBS-Tween and incubated with 1:4000 anti-SCCA1 antibody (NBP2, Novus International, Saint Louis, MO) overnight at 4 °C, 1:100,000 anti-Actin (A5441, Santa Cruz Biotechnology, Dallas, TX), 1:2000 anti-GAPDH-HRP-conjugated antibody (D16H11, Cell Signaling Technology, Danvers, MA) for 2 h at room temperature, 1:1000 anti-GSMDC1 (NBP2-33422, Novus Biologicals, LLC, Littleton, CO) overnight at 4 °C, 1:1000 anti-p-MLKL (91689 S, Cell Signaling Technology, Danvers, MA) overnight at 4 °C, 1:1000 anti-Cleaved Caspase-3 (9661, Cell Signaling Technology, Danvers, MA) overnight at 4 °C,, 1:1000 anti-Cleaved Caspase-7 (8438, Cell Signaling Technology, Danvers, MA) overnight at 4 °C, 1:1000 anti-Cleaved PARP (5625 s, Cell Signaling Technology, Danvers, MA) overnight at 4 °C, 1:1000 anti-Full Length PARP (9524 s, Cell Signaling Technology, Danvers, MA) overnight at 4 °C, 1:1000 anti-Caspase-7 (12827, Cell Signaling Technology, Danvers, MA) overnight at 4 °C, 1:500 anti-Caspase-3 (sc-7272, Santa Cruz biotechnology, INC, Texas) overnight at 4 °C, 1:1000 anti-RIP3 (57220, Cell Signaling Technology, Danvers, MA) overnight at 4 °C, 1:1000 anti-RIPK3 (ser227) (93654 s, Cell Signaling Technology, Danvers, MA) overnight at 4 °C, 1:1000 anti-RIPK1 (ser166) (44590, Cell Signaling Technology, Danvers, MA) overnight at 4 °C, 1:10000 anti-p-ATM

(ab81292-100UL, Abcam, Cambridge, MA) overnight at 4 °C, 1:1000 anti-ATM (NB100-309, Novus Biologicals, LLC, Littleton, CO) overnight at 4 °C, 1:500 anti-cathepsin L-HRP (sc-32320, Santa Cruz Biotechnology, INC, Texas), 1:2000 anti-cathepsin S (AF1183, R&D Systems, Minneapolis, MN), and 1:1000 anti-cathepsin K (sc-48353, Santa Crus Biotechnology, INC, Texas). Anti-mouse or anti-rabbit HRP-conjugated secondary antibody was used for detection with ECL chemilu-minescent reagent (GE Healthcare Life Sciences, Pittsburgh, PA), visualized, and quantified using the Bio-Rad ChemiDoc MP imaging system and Image Lab software (Bio-Rad, Hercules, CA).

**Clonogenic cell survival assay**. 500–1000 cells per well were seeded in 6-well plates 24 h prior to treatment with increasing doses of radiation (2, 4, 6 Gy x 1) and incubated for 1–3 weeks until control plates formed visible colonies (≥50 cells). IC50 concentration of cisplatin was added 1 h prior to radiation, where indicated. Plates were fixed and stained with 0.5% Crystal Violet, 30% Methanol, 10% Acetic Acid, 60% ddH2O for 30 min, rinsed in tap water and air dried at room tem-perature. Surviving fraction was calculated as the number of colonies ÷ (500 * plating efficiency) and plotted on a $\log_{10}$ scale as per convention. The linear quadratic equation was fit to each dataset using GraphPad Prism 8©. The dose modifying factor (DMF) was determined as the ratio of the dose resulting in 10% surviving fraction compared to control, indicated as the reference condition.

**Cell Titer-Glo survival assay**. Cells were seeded to 50–70% confluence in 96-well dishes and cultured overnight prior to treatment with increasing doses of ionizing radiation. 24 h later cells were washed once with PBS on ice then lysed directly in the well using Cell Titer-Glo reagent and incubated per manufacturer's protocol prior to measurement of luminescence on the SpectraMax i3 plate reader (Mole-cular Devices, San Jose, CA). Mean relative light units (RLUs) normalized to background value of a well with no cells was plotted with standard deviation.

**Time-lapse confocal microscopy**. Cells were seeded as described in the cell death assay except on Nunc™ Lab-Tek™ II 8-well Chambered Coverglass (155409PK, Thermo Scientific™) or 35 mm coverglass culture dishes (MatTek Life Sciences, P35G-1.5-14-C), irradiated ~24 h after seeding and placed on the Leica TCS SP8 X confocal microscope (Leica Microsystems Inc., Buffalo Grove, IL) beginning 24, 48, 72, or 96 h after IR and imaged every 20–30 min using a ×63/1.4 N.A. oil objective for ~16 h at a time at 37 °C in 5% CO2 using an OKO-labs cage incubator. Prior to imaging, cells were stained with LysoTracker™ Deep Red (Invitrogen™, Life Tech-nologies Corp., Carlsbad, CA) for 15–30 min, washed twice with PBS and stained with propidium iodide (1:500) in FluoroBrite DMEM media (Invitrogen, Life Technologies Corp., Carlsbad, CA) with 10% FBS and 4mM L-glutamine. Images were collected using LASX software (Leica Microsystems, Buffalo Grove, IL) and visualized and analyzed using Volocity software (v6.3.5, Quorum). Image contrast and brightness were adjusted for presented images in Microsoft PowerPoint with similar adjustments for all images in the experiment (acquired with identical microscope settings). As acquired images available for review.

**Transmission electron microscopy**. Cells were plated in 60 mm dishes to reach a confluence of ~70% at the time of RT, then treated 24 h after plating with sham, 10 Gy (HT3) or 30 Gy (SW756) RT. Cells were harvested by trypsinization 96 h after treatment. For ultrastructural analyses, samples were fixed in 2% paraf-ormaldehyde/2.5% glutaraldehyde (Polysciences Inc., Warrington, PA) in 100 mM sodium cacodylate buffer, pH 7.2 for 1 h at room temperature. Samples were washed in sodium cacodylate buffer at room temperature and postfixed in 1% osmium tetroxide (Polysciences Inc.) for 1 h. Samples were then rinsed extensively in dH2O prior to en bloc staining with 1% aqueous uranyl acetate (Ted Pella Inc., Redding, CA) for 1 h. Following several rinses in dH2O, samples were dehydrated in a graded series of ethanol and embedded in Eponate 12 resin (Ted Pella Inc.). Sections of 95 nm were cut with a Leica Ultracut UCT ultramicrotome (Leica Microsystems Inc., Bannockburn, IL, USA), stained with uranyl acetate and lead citrate, and viewed on a JEOL 1200 EX transmission electron microscope (JEOL USA Inc., Peabody, MA, USA) equipped with an AMT 8 megapixel digital camera and AMT Image Capture Engine V602 software (Advanced Microscopy Techni-ques, Woburn, MA).

**Lactate dehydrogenase (LDH) enzyme activity assay**. Cells were seeded to 50–75% confluence in 96-well dishes and cultured overnight prior to treatment with ionizing radiation. Spontaneous release of lactate dehydrogenase (LDH) was conducted at 24 h time points following the kit protocol of the Cytotoxicity Detection Kit (Roche, Indianapolis, Indiana, USA). Absorbance at 490 nm and 680 nm wavelengths was measured on the SpectraMax i3 plate reader (Molecular Devices, San Jose, CA).

**In vivo tumor growth and radiation response**. Female athymic nude mice were inoculated at 7–10 weeks of age with $0.5 \times 10^6$ cells suspended in a 1:1 mixture of IMDM media and Matrigel® Matrix (Corning, Glendale, AZ, USA) subcutaneously in the flank region. Tumor growth was measured by caliper twice weekly beginning at 3 days post-inoculation to determine estimated tumor volume assuming an

ellipsoid sphere (V = 0.5*(L × W^2)), where V = volume, L = length of the long-axis, W = width of the short axis. Once tumors reached an appropriate size, ani-mals were randomized to treatment with indicated doses of IR or sham IR on the SARRP as described in Irradiation of cell lines and mice section. Following monitored recovery from anesthesia, tumor growth was measured by caliper twice weekly until tumors reached a maximal dimension of 20 mm or otherwise reached a pre-defined sacking criterion of skin ulcer.

**Tumor tissue analysis by histology and immunohistochemistry**. At tumor harvest, mice were sacrificed using a CO2 asphyxiation chamber and tumors immediately harvested and divided into three components for flash freezing, into 4% paraformaldehyde (PFA), and fixation buffer for TEM as described above. PFA-fixed tumor was sent and histology was performed by HistoWiz Inc. (histowiz.com) using a Standard Operating Procedure and fully automated workflow. Samples were processed, embedded in paraffin, and sectioned at 4µm. Immunohistochemistry was performed on a Bond Rx autostainer (Leica Bio-systems) using standard protocols. Antigen retrieval method was heat induced epitope retrieval (HIER) at pH = 6.0 for 20 min (CC-3) and enzyme digestion for 10 min (TUNEL). Antibodies used were anti-cleaved-caspase-3 (Asp175) (Cell Signaling, 9661 S, 1:300) and rabbit anti-rat secondary (Vector, 1:100). Bond Polymer Refine Detection Kit (Leica Biosystems, DS9800) was used according to manufacturer's protocol. After staining, sections were dehydrated and film coverslipped using a TissueTek-Prisma and Coverslipper (Sakura). Whole slide scanning (×40) was performed on an Aperio AT2 (Leica Biosystems). TUNEL staining was performed to determine TUNEL positive nuclei as a measure of fragmented nuclear DNA under standard conditions using the Promega Dead-End Fluorometric Detection System (Promega, G3250). QuPath V0.1.2 open source software was used for digital image viewing and automated analysis using macros for nuclear staining (TUNEL) and cytoplasmic staining (CC-3) to determine percent cells positive. Positive control slides processed in batch under identical conditions were used to optimize macros. Full digital slide images are available at https://app.histowiz.com/shared_orders/2ccb0ce8-e4e1-41d1-8f75-a0e621ced36e/slides/.

**Whole-transcriptome RNA-sequencing (RNA-seq) analysis of lysosomal cysteine cathepsins**. Patients enrolled on a prospective tumor banking study with written informed consent (Washington University IRB No. 201105374) submitted pre-treatment tumor biopsies which were cryopreserved for future analysis. Tumor samples with sufficient high-quality RNA for whole-transcriptome sequencing (RNA-seq), as defined by the criteria utilized for The Cancer Genome Atlas (TCGA), were included in this study, $n = 68$ patients[36]. Patient and treatment characteristics of this cohort have been previously published[37,38], and the full RNA-seq datasets are available on Gene Expression Omnibus (GEO): (https://www.ncbi.nlm.nih.gov/geo/query/acc.cgi?acc=GSE151666). This analysis of RNA-seq data were approved by the Washington University IRB under protocol number 201710187.

**Statistics and reproducibility**. For animal experiments, based on our published and preliminary in vitro data, using a Bonferroni correction with 0.05 significance, 80% power and a two-sided $t$-test, we estimated needing 4 mice per group to determine an effect size of 3.5 for tumor growth kinetics. Given an anticipated tumor-take rate of 80% for HT3 and SW756 cell lines, we injected 5 mice per group for those experiments. Because all of the mice injected developed tumors, we included all in the experiment and analysis. For C33A cell line, tumor-take rate was closer to 100% so 4 mice per group were injected. Only three mice in the C33A group developed tumors. Tumor volume was estimated based on the two-dimensional caliper measurements and equation for the volume of an ellipsoid sphere: V = 0.5 * (L × W2), and means were plotted with standard error for the group. Two-way ANOVA with Bonferroni correction was performed to determine difference between mean tumor growth curves. Students' $t$-test was used to com-pare groups in flow cytometry, cell death, TEM quantification and histology (CC-3 and TUNEL) assays, unless where otherwise indicated. Individual clonogenic survival data points were fit with the linear quadratic equation, and dose modifying factors (DMF) were calculated at 10% surviving fraction. In all cases, $p$-value of <0.05 was considered significant. All cell-line-based experiments were performed in triplicate with at least three biologic replicates, except as noted in the figure legends.

**Reporting summary**. Further information on research design is available in the Nature Research Reporting Summary linked to this article.

## Data availability

All raw data included in this manuscript including but not limited to source data underlying all graphs and charts, raw blot images, live-cell image videos, and TEM images are available at: https://data.mendeley.com/datasets/8465mbnxt7/1. Full IHC slide images are available at: https://app.histowiz.com/shared_orders/2ccb0ce8-e4e1-41d1-8f75-a0e621ced36e/slides/. RNA-seq transcriptomic data are available on Gene Expression Omnibus (GEO): accession number: GSE151666. Plasmid backbones are available on addgene (#52961, #24129) and Takara Bio USA (631334). Plasmids and

sequences with gene of interest inserts are available upon request by contacting the corresponding author.

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

## Acknowledgements

This work was supported by research grants from the Elsa U Pardee Foundation, American Society for Clinical Oncology (ASCO) Career Development Award, American Society for Therapeutic Radiation Oncology (ASTRO) Junior Faculty Award, the National Cancer Institute (NIH K08CA237822 and K12 CA167540 to SM), the National Institute of Diabetes and Digestive and Kidney Diseases (NIH R01DK104946 (to GAS), R01DK114047 (to CJL, GAS), and The Children's Discovery Institute of St. Louis Children's Hospital Foundation (to CJL, SCP). The content is solely the responsibility of the authors and does not necessarily represent the official views of the NIH. We would like to acknowledge Wandy Beatty in the Molecular Microbiology Imaging Facility, for TEM processing and expert assistance in image acquisition and analysis. TEM experiments were performed in part through the use of Washington University Center for Cellular Imaging (WUCCI) supported by Washington University School of Medicine, The Children's Discovery Institute of Washington University and St. Louis Children's Hospital (CDI-CORE-2015-505 and CDI-CORE-2019-813) and the Foundation for Barnes-Jewish Hospital (3770 and 4642). We would also like to thank Dr. Cedric Mpoy and Dr. Buck Rogers of the Small Animal Radiation Research Platform Core, as well as Michael Zahner for his assistance with animal experiments, Dr. Xiaowei Wang for his design of the WU-CRIPSR gRNA Designer which was used for design of the SERPINB3-gRNA, Dr. Ekkehard Weber (Martin Luther Univ. Germany) for sharing the monoclonal anti-cathepsin B antibody, and Lena Zein for administrative assistance. We would also like to thank members of the Washington University Histology Core in the Developmental Biology Department for processing of mouse tumor tissue.

## Author contributions

S.W., C.J.L., G.A.S., P.W.G., and S.M. made substantial contributions to the conception, design, acquisition, analysis and interpretation of the data, drafted and substantively revised the work. S.C.P., A.A.A., and J.K.S. have contributed to the conception and substantively revised the work. V.S., L.C., J.M., A.A., K.J., J.Z., Y.H., and M.P. have contributed substantially to the acquisition, analysis and interpretation of the data, and substantively revised the work. All authors have reviewed and approved the submitted version of the manuscript and are accountable for their own contributions to the manuscript.

## Competing interests

The authors declare no competing interests.

**Additional information**

