## [Peer Review File · Communications Biology]

Reviewers' comments:

Reviewer #1 (Remarks to the Author):

The manuscript by Wang and coworkers entitled "The squamous cell carcinoma antigen/SERPINB3 protects cervical cancer cells from chemoradiation by preventing lysoptosis" is a first report to describe the cytoprotective role of SERPINB3 against radiation-induced necrosis. Authors provided strong evidences that cells (and mice) lacking SERPINB3 protein are more sensitive to radiation and cisplatin-induced cell death. Next, authors demonstrated that the cell death induced by radiation in SERPINB3-lacking cells is lysoptosis. By using a panel of cell death specific inhibitors authors excluded other cell death types such as pyroptosis, necroptosis, apoptosis and ferroptosis. Overall, the manuscript is well written and easy to follow. The Materials and Methods section is described in details, and the experiments performed fully support the research hypothesis that SERPINB3 serves as a radioprotective factor in cervical cancer cells by regulating lysoptosis, a lysosome-mediated necrosis cell death. Below there are some concerns that need to be addressed prior to publication.

1. Authors do not explain what is the main protease target for SERPINB3 in cervical cancer cells (is it one enzyme, or rather a group of proteases). In the Reviewer`s opinion this is key information to fully understand the mechanism of lysoptosis. In this paper authors only performed the experiments with E64d which is a general inhibitor of cysteine cathepsins, with no further investigation of individual enzymes from this family.

2. The lysoptosis is a generally new concept in the field of cell death, therefore this manuscript can definitely reach the broad audience. However, in the manuscript authors mentioned that another paper on the lysoptosis (from the same group, by Good et al.) is pending for publication in Communications Biology. Therefore it is difficult to evaluate whether the present manuscript provides a conceptual breakthrough, or is a follow-up and incremental study that strongly relies on previous findings.

Reviewer #2 (Remarks to the Author):

Wang et al. explore the mechanism by which SERPINB3 protects cervical cancer cells from ionizing radiation (IR)-induced cell death. The authors show that SERPINB3 KO cervical cancer cells are more sensitive to IR-induced cell death in vitro and that doubling time was higher in vivo for SERPINB3 KO tumors. The authors also determine that changes to cell survival in the SERPINB3 KO cells is not a function of changes in cell cycle distribution or compromised DNA repair. Further, the mechanism of cell death in SERPINB3 KO cells was determined to be dependent on lysosomal permeabilization. The authors also created a cell line with mutant SERPINB3 that was deficient in protease-inhibitory function which was sufficient to sensitize cells to IR-induced cell death. The novelty of these data is in the mechanism of cell death as other studies have primarily shown that SERPINB3 protects cancer cells from apoptosis using other models (e.g. Suminami et al., British Journal of Cancer 2000). While these data are relevant to the cell death and tumor biology field, the following revisions should be addressed prior to publication.

1. The authors use the term lysoptosis throughout the paper. However, the term lysoptosis does not seem to be previously defined. Is this referring to lysosome-dependent cell death? If not, please explain the difference. Otherwise, please refer to Galluzzi et al 2018 (<https://doi.org/10.1038/s41418-017-0012-4>) for proper naming of your cell death pathway.

2. Figure 1: The authors use both KO alone and B3-KO to label the panels in this figure. Are these the same cells?

3. Figure 1 and throughout: The figures would be much easier to read with legends that clearly labeled the control group. The author's use of 'c' as a shorthand for control was also not defined in the main text.

4. Figure 1: It seems that cisplatin has no effect in the HT3 cells and little effect in the SW756

cells. Was this expected?

5. Supplemental Figure 1C-1F: Tumor growth should be displayed on a log scale and the scales should be standardized. The authors conclude in the text that there is delayed tumor growth in the KOs, but this is not evident from the way these graphs are presented.

6. Growth curves in Supplemental Figure 1 and Supplemental Figure 4 should be included as part of the main text.

7. Supplemental 1E and 1F: Are these the same parental cell lines (SW756 and SW857)? If not, the authors should explain why they are being compared.

8. Line 183-184: "Although some absolute differences were significant, we found no meaningful differences in cell cycle distribution either de novo or following 4Gy to explain the differential radiosensitivity (Figure 3A-D)." The authors should present their conclusions more objectively instead of stating what may or may not be meaningful.

9. Figure 4: A functional readout for necrosis such as an LDH release assay would be beneficial to reinforce the conclusions that these cells are necrotic.

10. Line 219: The heading for this section states that cell death is primarily necrotic, but the authors show necrotic morphology with changes in apoptotic proteins. The authors should be careful about what they conclude from this section. Writing that B3-KO cells show characteristics of multiple cell death mechanisms may be more appropriate.

11. Figure 7: A functional validation of the A341R cells should be shown here or in supplemental.

12. Supplemental Figure 4 is never referred to in the text.

13. The authors should provide information in the figure legend on the number of data points or repeats for each figure panel.

Reviewer #3 (Remarks to the Author):

In this work by Wang et al., the authors studied the protective mechanism of endogenous lysosomal cysteine protease inhibitor SERPINB, which is elevated in patients with cervical cancer, against ionizing radiation (IR). They identified that SERPINB3 protects against IR by inhibiting lysosome leakage mediated necrosis. They showed knock out of SERPINB3 sensitizes to IR, while its expression can cause resistance to IR. Growing evidence shows that IR can cause mixture of cellular outcomes including mitotic catastrophe, iron dependent cell death, senescence... (see the "Adjemian et al. Cell Death and Disease (2020)11:1003 <https://doi.org/10.1038/s41419-020-03209-y>"). The authors also observed activation of apoptosis upon irradiation, at least in one of the cell line, in addition to lysosomal leakage, which could also point toward the different mixture of events in IR induced cell death. The authors performed well-designed experiment, however, based on their results it seems that lysosomal leakage is not the only type of cell death induced by IR. So, I would not consider lysosomal leakage as the only type of cell death. To support the role of lysosomes in cell death, they could also check iron chelators to see the cell death could be inhibited or not.

Figure 1 and its related results:

It would be good if the authors provide the western blot image showing the knock-out of SERPINB3. Did the authors use a single clone of KO in their further experiment?

Could the authors indicate the type of statistical tests and the n of biological replicates?

If the author considers the mean of tumor volume of B3KO and C groups, could they draw the same conclusion that B3KO group have a less tumor volume? Or is it just delay in the growth?

The radiation time in SW756-C is different from B3 KO cells, because of the slow tumor growth rate. Could also the effect that is seen on the time of doubling size be related to the slow tumor

growth rate?

Could the author provide IHC on the tumor samples from mice? To show there is more cell death in B3KO tumors?

Fig 5 and its related results:

Could the author quantify their WB results? It seems there is more Bcl2 and less BAX in SW756 KO cells.

Although Fer-1 is a potent ferroptosis inhibitor with IC50 value below 20nM, it is recommended that the authors use high concentrations of Fer-1 (1uM-10uM) as well in their experiment. Also they could use iron chelator to see whether cell death can be inhibited or not.

The representative IHC staining does not match with the quantified data for cleaved caspase 3 and TUNEL. It seems there is more cleaved caspase 3 and TUNEL positivity in KO samples. Which would make sense if the tumor growth is less in KO tumors.

It is suggested that the authors check lipid peroxidation in cells as well by using c-11 Bodipy, to see they have lipid peroxidation or not and whether it can be affected upon SERPINB3 KO.

The authors would like to thank the reviewers for their thorough and thoughtful review of our manuscript. Below is a point-by-point response to each of the three reviewers' comments, with reference to specific revisions in the current version.

Reviewers' comments:

“Reviewer #1 (Remarks to the Author):

The manuscript by Wang and coworkers entitled "The squamous cell carcinoma antigen/SERPINB3 protects cervical cancer cells from chemoradiation by preventing lysoptosis" is a first report to describe the cytoprotective role of SERPINB3 against radiation-induced necrosis. Authors provided strong evidences that cells (and mice) lacking SERPINB3 protein are more sensitive to radiation and cisplatin-induced cell death. Next, authors demonstrated that the cell death induced by radiation in SERPINB3-lacking cells is lysoptosis. By using a panel of cell death specific inhibitors authors excluded other cell death types such as pyroptosis, necroptosis, apoptosis and ferroptosis. Overall, the manuscript is well written and easy to follow. The Materials and Methods section is described in details, and the experiments performed fully support the research hypothesis that SERPINB3 serves as a radioprotective factor in cervical cancer cells by regulating lysoptosis, a lysosome-mediated necrosis cell death. Below there are some concerns that need to be addressed prior to publication.

1. Authors do not explain what is the main protease target for SERPINB3 in cervical cancer cells (is it one enzyme, or rather a group of proteases). In the Reviewer`s opinion this is key information to fully understand the mechanism of lysoptosis. In this paper authors only performed the experiments with E64d which is a general inhibitor of cysteine cathepsins, with no further investigation of individual enzymes from this family.”

Response: We thank the reviewers for raising this point and are also interested in understanding the downstream effectors/executors of lysoptosis in the context of radiation-induced tumor cell death, namely which one(s) of the cysteine proteases are required and/or sufficient. As the reviewer points out, available pharmacologic inhibitors are not specific for any one cysteine protease (though E64/E64d are felt to be the most specific for lysosomal cysteine proteases), and available protease substrate reagents offer only a similar level of specificity. Therefore, we instead used a genetic approach to investigate which protease(s) is/are involved. Since submission of this manuscript we have determined expression levels of the lysosomal cysteine cathepsins in primary cervix tumors (Figure 7H). We find expression of cathepsin B (CTSB), cathepsin H (CTSH), cathepsin C (CTSC), and cathepsin X (CTSX), also known as cathepsin Z, and lower levels of cathepsin L (CTSL), cathepsin S (CTSS), and cathepsin K (CTSK). We find very low or undetectable levels of cathepsin V (CTSV), and cathepsin F (CTSF). CTSB and CTSH are primarily exopeptidases at acidic pH, and CTSX and CTSC are exclusive exopeptidases and therefore unlikely to be inhibited by the pseudo-substrate bait and trap mechanism of SERPINB3. On assessment of the SW756 cell line, CTSL was detectable, while neither CTSK or CSTS were detectable by WB (Figure 7I and Supplemental Figure 6). Therefore, we generated stable knock-out cell lines of *CTSL* in the SW756-B3-WT and

-B3-KO backgrounds (Figure 7I). Cell death assays are shown in Figure 7J and show partial rescue of radiation-induced death in the B3-KO / CTSL-KO cell line, demonstrating that CTSL is at least one of the protease targets of SERPINB3 and is responsible for enhanced radiation sensitivity in this system. This is consistent with the *in vitro* data that SERPINB3 is a potent inhibitor of cathepsin L at 1:1 stoichiometry (Schick et al, Biochemistry. 1998 Apr 14;37(15):5258-66. doi: 10.1021/bi972521d). Cathepsin proteases are a complex system of auto- and trans-activating proteases, and this upstream signaling is the focus of ongoing study in the lab.

“2. The lysoptosis is a generally new concept in the field of cell death, therefore this manuscript can definitely reach the broad audience. However, in the manuscript authors mentioned that another paper on the lysoptosis (from the same group, by Good et al.) is pending for publication in Communications Biology. Therefore it is difficult to evaluate whether the present manuscript provides a conceptual breakthrough, or is a follow-up and incremental study that strongly relies on previous findings.”

Response: The reviewer is correct, that lysoptosis is a newly identified and termed cell death mechanism that is defined as distinct in a companion manuscript previously submitted to Communications Biology and provided for review as an attachment to the initial submission of this manuscript. We believe this manuscript provides additional conceptual breakthrough for several reasons: 1) mechanisms of radiation-induced cell death in solid tumors, and especially in cervical cancer, are poorly understood. This provides the first evidence that a lysosome-mediated cell death mechanism occurs in response to clinically-relevant doses of radiation. 2) SERPINB3 is upregulated in many cancer types, and in addition to the potential growth advantage and impact on chemotherapy-induced apoptosis (accounting for a small percentage of death), we show here that a major implication of SERPINB3 overexpression is that it functions to protect tumor cells from radiation-induced death. 3) We show that the reactive site loop is necessary for protection against radiation-induced death both *in vitro* and *in vivo* (not addressed in the Good, Markovina *et al* manuscript), demonstrating molecular mechanism and also suggesting a potential therapeutic target to achieve improved radiosensitivity. Finally, thanks to the reviewer’s previous suggestion, we have identified cathepsin L as at least one protease target of SERPINB3 that mediates radiation-induced cell death in the absence of B3. The findings presented in this manuscript at last provide the rationale to support development of a SERPINB3-targeting therapeutic, which will enhance radiosensitivity and tumor control in patients with high SERPINB3-cervix cancers. This would not only improve recurrence but has the potential to finally improve cancer-specific survival, which has not changed for this disease in nearly 50 years.

“Reviewer #2 (Remarks to the Author):

Wang et al. explore the mechanism by which SERPINB3 protects cervical cancer cells from ionizing radiation (IR)-induced cell death. The authors show that SERPINB3 KO cervical cancer cells are more sensitive to IR-induced cell death *in vitro* and that doubling time was higher in

vivo for SERPINB3 KO tumors. The authors also determine that changes to cell survival in the SERPINB3 KO cells is not a function of changes in cell cycle distribution or compromised DNA repair. Further, the mechanism of cell death in SERPINB3 KO cells was determined to be dependent on lysosomal permeabilization. The authors also created a cell line with mutant SERPINB3 that was deficient in protease-inhibitory function which was sufficient to sensitize cells to IR-induced cell death. The novelty of these data is in the mechanism of cell death as other studies have primarily shown that SERPINB3 protects cancer cells from apoptosis using other models (e.g. Suminami et al., British Journal of Cancer 2000). While these data are relevant to the cell death and tumor biology field, the following revisions should be addressed prior to publication.

1. The authors use the term lysoptosis throughout the paper. However, the term lysoptosis does not seem to be previously defined. Is this referring to lysosome-dependent cell death? If not, please explain the difference. Otherwise, please refer to Galluzzi et al 2018 (<https://doi.org/10.1038/s41418-017-0012-4>) for proper naming of your cell death pathway.”

Response: The reviewer is correct, lysoptosis is a newly identified and termed cell death mechanism defined in a companion manuscript previously submitted to Communications Biology, and provided for review as an attachment to the initial submission of this manuscript. Lysosomal membrane permeabilization (LMP) and cathepsin release are the hallmarks of lysosome-dependent cell death (LDCD). However, LMP is detected in most regulated cell death programs suggesting LDCD is not an independent cell death pathway, but rather a process conscripted by other cell death routines to assist in the final demise of the cell. In *Caenorhabditis elegans* (*C. elegans*) null for the intracellular serpin and lysosomal cysteine protease inhibitor, SRP-6 (homologue of Serpinb3), animals exposed to different types of stress undergo a stereotypical LDCD pathway characterized sequentially by a rise in intracellular calcium, calpain activation, a rapid loss of lysosomal membrane integrity and a wave of lysosomal cathepsin-dependent cytoplasmic proteolysis. Death is independent of caspases and *C. elegans* lacks the lytic machinery required to execute necroptosis and pyroptosis (Luke *et al*, Cell. 2007 Sep 21;130(6):1108-19. doi: 10.1016/j.cell.2007.07.013). We have designated this cell death routine, lysoptosis, to distinguish it from other conditions employing LMP and provide strong evidence that this pathway is conserved in higher order metazoans (mice and humans). The SERPINB3-KO tumor lines described in this manuscript were used in the accompanying manuscript (Good *et al*) and show that they are primed for lysoptosis, even if apoptosis, ferroptosis, and necroptosis triggers are applied. Moreover, none of the standard inhibitors of apoptosis, necroptosis, pyroptosis, ferroptosis, MOMP block lysoptosis. This manuscript provides evidence that ionizing radiation is one (highly clinically relevant) upstream stressor that can induce lysoptosis in the absence of the protective SERPINB3. Based on the editors’ preference and outcomes of both manuscripts (with respect to acceptance for publication and timing), we can either include specific reference to the Good *et al* paper, or not.

“2. Figure 1: The authors use both KO alone and B3-KO to label the panels in this figure. Are these the same cells?”

Response: They are the same cells, and this inconsistency has been corrected – all B3-KO cells are now labeled as such.

“3. Figure 1 and throughout: The figures would be much easier to read with legends that clearly labeled the control group. The author’s use of ‘c’ as a shorthand for control was also not defined in the main text.”

Response: This has been clarified in Figure 1 and throughout the text/figures to be consistent. The parental cell line expressing the CRISPR-Cas9 vector with no gRNA (CRISPR-Control) is now referred to as “B3-WT.” Definitions are provided in the methods section and in the second paragraph of the results section, when these lines are first introduced.

“4. Figure 1: It seems that cisplatin has no effect in the HT3 cells and little effect in the SW756 cells. Was this expected?”

Response: The authors raise an interesting point that we did not emphasize in the current manuscript. Despite the generally accepted thought that cisplatin is a radiosensitizer, we and others have not been able to demonstrate significant radiosensitizing effect of cisplatin *in vitro* with cervix cancer cell lines using logically-selected concentrations (Britten et al, Int J Radiat Oncol Biol Phys. 1996 Jan 15;34(2):367-74. doi: 10.1016/0360-3016(95)02088-8). For this study, we determined the single-agent IC50 for cisplatin in both the SW756 and HT3 parental cell lines and used these concentrations for combination therapy, again confirming minimal radiosensitizing effect of cisplatin. Plating efficiencies are lower in the cisplatin-treated wells, but this does not translate to a lower surviving fraction. Interestingly, our group has recently published a study in which intra-tumoral platinum levels were measured using mass spectrometry from tumor biopsies taken from women undergoing definitive chemoradiation therapy for cervical cancer. In this study, we showed that at the time of radiation (when biopsies were taken), the intra-tumoral concentration of cisplatin was on average 5-fold lower in the tumor biopsies compared to cell line IC50 (Federico C et al, “Localized Delivery of Cisplatin to Cervical Cancer Improves Its Therapeutic Efficacy and Minimizes Its Side Effect Profile.” Int J Radiat Oncol Biol Phys. 2021 Apr 1;109(5):1483-1494. doi: 10.1016/j.ijrobp.2020.11.052. PMID: 33253820). Thus, it is possible that *in vivo* effects of cisplatin when given intravenously are even less effective at radiosensitization.

In the revised version, we have included the plating efficiencies for experiments shown in Fig 1F, G as part of Supplemental Figure 1C, D, and included a line in the Discussion, including citations of the above referenced manuscripts.

“5. Supplemental Figure 1C-1F: Tumor growth should be displayed on a log scale and the scales

should be standardized. The authors conclude in the text that there is delayed tumor growth in the KOs, but this is not evident from the way these graphs are presented.”

Response: Thank you for this suggestion, the spider plots in SF 1C-F have been replaced by graphing mean +/- SEM tumor volumes from the time of RT on a logarithmic scale. These figures are now part of Figure 2, as well as the calculated doubling time values for each individual tumor (doubling time = $\ln 2 / (\% \text{ change volume} / \text{day})$) is also included in Figure 2 for both cell line *in vivo* experiments. Kaplan Meier curves were moved to Supplemental Figure 1.

“6. Growth curves in Supplemental Figure 1 and Supplemental Figure 4 should be included as part of the main text.”

Response: As per response to #5 above, the growth curves are now presented in Figure 2 and described in the main text.

“7. Supplemental 1E and 1F: Are these the same parental cell lines (SW756 and SW857)? If not, the authors should explain why they are being compared.”

Response: Thank you for noting this typo which has been corrected, both are SW756 cells.

“8. Line 183-184: “Although some absolute differences were significant, we found no meaningful differences in cell cycle distribution either de novo or following 4Gy to explain the differential radiosensitivity (Figure 3A-D).” The authors should present their conclusions more objectively instead of stating what may or may not be meaningful.”

Response: As is appropriate, subjective interpretation of the data was removed from the text, leaving only objective description of the salient results. One sentence describing our interpretation of the data is now included in the Discussion, paragraph 5.

“9. Figure 4: A functional readout for necrosis such as an LDH release assay would be beneficial to reinforce the conclusions that these cells are necrotic.”

Response: We agree with the reviewer that a quantitative measure of necrotic cell death mechanism would be useful. However, as the reviewers know, LDH release into the media results from plasma membrane permeability that is not specific to the mode of cell death. The Sytox reagent assays performed and presented in the original manuscript reflect the same non-specific process (plasma membrane permeability) and is also quantitative and we find more reliable than the LDH assay, as it relies on imaging of the undisturbed cells and does not require manipulation, enzyme activity, etc. To our knowledge there is no biochemical assay that will corroborate the gold-standard transmission electron microscopy method of confirming necrotic mode of death. That said, per the reviewer’s

request, we did perform an LDH release assay after radiation and found that release of LDH from the cell was observed on a time course tracking with % Sytox positive nuclei. Some of these data are presented in Supplemental Figure 1G, and is described in the methods and results sections.

“10. Line 219: The heading for this section states that cell death is primarily necrotic, but the authors show necrotic morphology with changes in apoptotic proteins. The authors should be careful about what they conclude from this section. Writing that B3-KO cells show characteristics of multiple cell death mechanisms may be more appropriate.”

Response: We agree with the reviewers that while most evidence supports primarily necrotic cell death in B3-KO cells, there is evidence of some caspase cleavage/dependence in the HT3 background (not in the SW756 cell line). Thus, the heading of this section has been modified to: “Cell death in B3-KO cells following RT suggests potential engagement of multiple cell death mechanisms with primarily necrotic morphology.”

“11. Figure 7: A functional validation of the A341R cells should be shown here or in supplemental.”

Response: The SERPINB3-A341R mutation is a well-established loss of function mutation characterized and previously published by some of the co-authors of this manuscript (Schick et al, Proc Natl Acad Sci U S A. 1998 Nov 10; 95(23): 13465–13470). For the purposes of this manuscript, we generated the A341R mutant using site-directed mutagenesis and confirmed the desired mutation using Sanger sequencing. We demonstrated a protein of the proper size and selected a clonal line with equivalent protein levels to clonal lines expressing the wild-type SERPINB3 construct (Figure 7A). In the current data we are showing that although the protein is present in equivalent proportions and otherwise identical to the wild-type SERPINB3, it is unable to protect cervix cells / tumors from radiation in vitro (Fig 7B, C) and in vivo (Fig 7D-G). In the revised manuscript we have emphasized that this particular RSL mutant is well-established. We are open to suggestions from the reviewers as to how further to characterize the function of this mutant in the cell system.

“12. Supplemental Figure 4 is never referred to in the text.”

Response: Thank you for pointing out this oversight, a reference to Supplemental Figure 4 in the last paragraph of the Results section.

“13. The authors should provide information in the figure legend on the number of data points or repeats for each figure panel.”

Response: The number of data points and biologic replicates performed are now stated in the figure legends for each panel.

“Reviewer #3 (Remarks to the Author):

In this work by Wang et al., the authors studied the protective mechanism of endogenous lysosomal cysteine protease inhibitor SERPINB3, which is elevated in patients with cervical cancer, against ionizing radiation (IR). They identified that SERPINB3 protects against IR by inhibiting lysosome leakage mediated necrosis. They showed knock out of SERPINB3 sensitizes to IR, while its expression can cause resistance to IR. Growing evidence shows that IR can cause mixture of cellular outcomes including mitotic catastrophe, iron dependent cell death, senescence... (see the “Adjemian et al. *Cell Death and Disease* (2020)11:1003 <https://doi.org/10.1038/s41419-020-03209-y”>). The authors also observed activation of apoptosis upon irradiation, at least in one of the cell line, in addition to lysosomal leakage, which could also point toward the different mixture of events in IR induced cell death. The authors performed well-designed experiment, however, based on their results it seems that lysosomal leakage is not the only type of cell death induced by IR. So, I would not consider lysosomal leakage as the only type of cell death. To support the role of lysosomes in cell death, they could also check iron chelators to see the cell death could be inhibited or not.”

Response: We agree with the reviewer that although the predominant mode of cell death in B3-KO cells is lysosome-mediated necrosis, there is evidence of at least some caspase-cleavage and caspase-dependent death (caspase-inhibitors provide partial protection), and have reframed our results and discussion section to emphasize this. The role of iron and lysosomal iron in particular in lysoptosis is unclear. While ferroptosis, a form of regulated necrosis that involves oxidative damage to lipids, does rely on intracellular iron, and can be inhibited by iron chelators, we do not observe inhibition of radiation-induced cell death in SERPINB3-KO cells with either iron chelators (now Supplemental Figure 4B), or the lipophilic antioxidant ferrostatin-1 (Figure 5F and dose response in Supplemental Figure 4A). As the critical event involved in lysoptosis is leakage of lysosomal enzymes and their activity in the cytoplasm, we do not think that the lack of protection with iron chelators means this is not a lysosome-dependent cell death pathway. The mechanism of lysosomal membrane permeabilization (LMP) in response to ionizing radiation (which may include lipid-peroxidation) is the subject of ongoing study in the lab.

“Figure 1 and its related results:

It would be good if the authors provide the western blot image showing the knock-out of SERPINB3.”

Response: We thank the reviewers for pointing out that although the verification of B3-KO in these lines is previously published, we had not included a confirmatory figure in this manuscript. WB showing knock-out in both lines is now shown in Figure 1A. Because the cathepsin L knock-out lines were derived from uniquely generated SW756-B3-WT and -B3-KO background lines (using Cas9-gRNA ribonucleoprotein complex), a separate Western blot is included in Figure 7I.

“Did the authors use a single clone of KO in their further experiment?”

Response: Single cell clone KO's were derived and used for these experiments. More than one was used to confirm the phenotype of radiation sensitivity; however, the majority of the figures/experiments were performed with one clonal cell line for each parental background. CRISPR-control cells were pooled cells. The added cathepsin-L knock out lines were derived from parental SW756 cells expressing Cas9 and a uniquely generated SW756-B3-KO line from Cas9-gRNA nucleoprotein complex and were analyzed as pool cells. This is described in the methods section and a Western blot showing levels of SERPINB3 and CTSL is shown in Figure 7I.

“Could the authors indicate the type of statistical tests and the n of biological replicates?”

Response: The statistical tests used and n of biologic and intra-experimental replicates are now detailed in the figure legends. Each experiment was repeated at least three times unless otherwise stated (as in the case of the *in vivo* animal experiments).

“If the author considers the mean of tumor volume of B3KO and C groups, could they draw the same conclusion that B3KO group have a less tumor volume? Or is it just delay in the growth?”

Response: The mean and standard errors of tumor growth in the *in vivo* experiments are now shown in Figure 2, and two-way ANOVA demonstrates no difference in growth rate between the sham treated B3-WT and B3-KO tumors. There is a significant difference between irradiated HT3-B3-WT and B3-KO tumors, and this difference does not reach significance in the SW756 tumor background. Given slow overall growth of the SW756 tumor line in athymic nude mice, and their radiation sensitivity with 10Gy, we were in the process of repeating this experiment with a dose response of radiation to determine if there is indeed a difference. However, this experiment had to be terminated upon the abrupt closure of our laboratory at the height of the COVID-19 pandemic, and unfortunately we do not have the funds to repeat it a third time.

“The radiation time in SW756-C is different from B3 KO cells, because of the slow tumor growth rate. Could also the effect that is seen on the time of doubling size be related to the slow tumor growth rate?”

Response: Thank you for raising this possibility. While SW756-B3-KO tumors did take longer to establish in the athymic nude flank compared to SW756-B3-WT tumors, the tumors grew at a similar rate once they reached the desired size for radiation (Figure 2D). We thank the reviewers for suggesting that tumor volume be plotted as mean +/- SEM on logarithmic scale, as this fact then became evident.

“Could the author provide IHC on the tumor samples from mice? To show there is more cell death in B3KO tumors?”

Response: We agree with the reviewers that to be able to quantify cell death in the mouse tumors would be ideal. We took a few steps to approach this. In addition to measuring

tumor growth longitudinally in the experiments shown in Figure 2, we also performed a short-term experiment, in which tumors were established in mice, and harvested 96 hours after irradiation (or sham) treatment, in order to detect and quantify some markers of cell death (Figure 5G-P). The 96 hour time point was selected as this was felt to be the most likely time to see radiation-induced cell death (as cell death would not be evident on the tumors harvested from the longitudinal experiment several weeks after a single fraction of radiation). However, there is no accepted histologic or immunohistochemical assay for cell death overall, and particularly for necrotic cell death. Therefore, we stained these tumors for cleaved caspase-3 (as a marker of caspase-dependent death) and with the TUNEL assay for fragmented nuclear DNA. Quantitation of these assays is shown in Figure 5, with representative images, and a link to view the full slides available in the methods section, and has now been added to the figure legend as well.

“Fig 5 and its related results:

Could the author quantify their WB results? It seems there is more Bcl2 and less BAX in SW756 KO cells.”

Response: The original figure has quantitation under the BCL-2 and BAX images – these were determined by normalizing the band intensity to GAPDH band intensity for that lane. For SW756, BCL2 normalized quant was 0.9 for B3-WT and 0.9 for B3-KO (sham, 0h), and 1.6 for B3-WT and 1 for B3-KO (10Gy, 72h). For BAX, relative quant for SW756-B3-WT was 0.9 versus 0.5 for B3-KO cells (sham, 0h), and 1.3 versus 0.6 for 10Gy, 72h. Therefore, the authors interpret these data as suggesting no difference in BCL-2 levels between B3-WT and B3-KO cells, and perhaps decreased levels of the pro-apoptotic protein BAX, suggesting possible downregulation of BAX-dependent apoptosis in this cell line. This hypothesis has not been tested as it is not a focus of the current study, rather these data support the premise of this paper that apoptosis is not a predominant mode of radiation-induced cell death in B3-KO cells.

“Although Fer-1 is a potent ferroptosis inhibitor with IC50 value below 20nM, it is recommended that the authors use high concentrations of Fer-1 (1uM-10uM) as well in their experiment. Also they could use iron chelator to see whether cell death can be inhibited or not.”

Response: As suggested by the reviewer, we have repeated experiments with B3-WT and B3-KO cells in both the HT3 and SW756 backgrounds to determine if higher concentrations can inhibit radiation-induced cell death. We find that while 50nM (as well as the higher concentrations) effectively inhibit erastin-induced cell death in SW756 parental cells, none of the concentrations up to 10uM inhibit radiation-induced death in any of the cell lines (Supplemental Figure 4A). Similarly, please see the response to this Reviewer’s first suggestion regarding iron chelators (results shown in Supplemental Figure 4B).

“The representative IHC staining does not match with the quantified data for cleaved caspase 3 and TUNEL. It seems there is more cleaved caspase 3 and TUNEL positivity in KO samples. Which would make sense if the tumor growth is less in KO tumors.”

Response: Thank you for this perspective. We have increased the size of the IHC images and replaced with different field of view of the slides perhaps more representative of the quantified data. Quantitation was performed on the whole tumor section (~7.5-20mm x 5-10mm area), and magnified images shown are ~20X magnification so a less-representative field of view was inadvertently selected. Different fields of view of the same slide are now shown for B3-KO treated tumor.

“It is suggested that the authors check lipid peroxidation in cells as well by using c-11 Bodipy, to see they have lipid peroxidation or not and whether it can be affected upon SERPINB3 KO.”

Response: We agree with the reviewer that the upstream signaling events of radiation-induced lysoptosis are of great interest. For instance, what event(s) mediate LMP – is it signaling from the nucleus in response to DNA-damage and activation of protease cascades? Is it direct damage to the lysosomal lipid membrane via lipid peroxidation? These investigations are ongoing in the lab, including the use of C-11 BODIPYTM to detect and quantify lipid peroxidation. However, since this reagent is not specific for lysosomal membrane, optimization of its use with other reagents to mark organelles (mitochondria, lysosomes) with proper quantitation over time after radiation is out of the scope of the current manuscript.

In addition to the requested changes detailed above, we have also added acknowledgments of funding sources, as well as two new co-authors, Dr. Jin Zhang, and Kay Ramachandran, who contributed to the data generation and analysis of the RNAseq data now a part of Figure 7, and also contributed to the revision of the manuscript figures and text.

REVIEWERS' COMMENTS:

Reviewer #1 (Remarks to the Author):

In the reviewer's opinion, the authors addressed all the concerns raised during the first round of the evaluation of the manuscript. The authors provided sufficient evidences for the major role of lysosomal cathepsin L (and it's counterpart SERPINB3 inhibitor) in radiation-induced cell death in solid tumors. Although the role of other lysosomal cathepsins has not been fully elucidated, the authors explained that cathepsins signaling if the focus of ongoing work. Therefore, I recommend this article to be published in Communications Biology.

Reviewer #2 (Remarks to the Author):

Wang et al. explore the mechanism by which SERPINB3 protects cervical cancer cells from ionizing radiation-induced cell death. The authors show that SERPINB3 KO cervical cancer cells are more sensitive to ionizing radiation-induced cell death in vitro and in vivo and that tumor doubling time was higher for SERPINB3 KO tumors. Interestingly, the authors determined that the mechanism of cell death in SERPINB3 KO cells is dependent on lysosomal permeabilization and creating mutant SERPINB3 cells deficient in protease-inhibitory function was sufficient to sensitize cells to IR-induced cell death. The novelty of these data is that the mechanism of cell death was lysosome-dependent, and this mechanism of cell death will be of interest to the broader scientific community.

All initial critiques of the paper were suitably addressed by the authors.

The only unknown that remains is how the scope of this paper compares to the initial, unpublished findings on lysoptosis in the Good et al manuscript.

Reviewer #3 (Remarks to the Author):

The author addressed my remarks and question, but still there are issues that I would like to draw their attention to:

The difference between lysosomal cell death and lysoptosis must be explained clearly. Based on the results of figure 5, they conclude that there is also role of ferroptosis, pyroptosis and necroptosis for IR induced cell death. But if we just look at the WB, we just see activation of apoptosis. Could their conclusion and title of the figure be adapted? The title should just refer to apoptosis and no other type of cell death

The authors would like to once again thank the reviewers for their thorough and thoughtful review of our manuscript. Below is a point-by-point response to each of the three reviewers' comments, with reference to specific revisions in the current version.

REVIEWERS' COMMENTS:

Reviewer #1 (Remarks to the Author):

In the reviewer's opinion, the authors addressed all the concerns raised during the first round of the evaluation of the manuscript. The authors provided sufficient evidences for the major role of lysosomal cathepsin L (and it's counterpart SERPINB3 inhibitor) in radiation-induced cell death in solid tumors. Although the role of other lysosomal cathepsins has not been fully elucidated, the authors explained that cathepsins signaling if the focus of ongoing work. Therefore, I recommend this article to be published in Communications Biology.

Reviewer #2 (Remarks to the Author):

Wang et al. explore the mechanism by which SERPINB3 protects cervical cancer cells from ionizing radiation-induced cell death. The authors show that SERPINB3 KO cervical cancer cells are more sensitive to ionizing radiation-induced cell death in vitro and in vivo and that tumor doubling time was higher for SERPINB3 KO tumors. Interestingly, the authors determined that the mechanism of cell death in SERPINB3 KO cells is dependent on lysosomal permeabilization and creating mutant SERPINB3 cells deficient in protease-inhibitory function was sufficient to sensitize cells to IR-induced cell death. The novelty of these data is that the mechanism of cell death was lysosome-dependent, and this mechanism of cell death will be of interest to the broader scientific community.

All initial critiques of the paper were suitably addressed by the authors.

The only unknown that remains is how the scope of this paper compares to the initial, unpublished findings on lysoptosis in the Good et al manuscript.

Response: We thank the reviewer again for careful consideration of the current manuscript and hope that they will have the opportunity to read the companion manuscript, which was provided for review, but we understand is a substantial undertaking to consider two draft manuscripts as a part of one review. We have included reference statements in the revised manuscript Introduction and Discussion sections briefly summarizing the findings of the Luke et al study and how it relates to – and is distinct from but builds upon – those findings.

Reviewer #3 (Remarks to the Author):

The author addressed my remarks and question, but still there are issues that I would like to draw

their attention to:

The difference between lysosomal cell death and lysoptosis must be explained clearly.

Based on the results of figure 5, they conclude that there is also role of ferroptosis, pyroptosis and necroptosis for IR induced cell death. But if we just look at the WB, we just see activation of apoptosis. Could their conclusion and title of the figure be adapted? The title should just refer to apoptosis and no other type of cell death

Response: Thank you for pointing out the remaining data requiring further clarification. We have included in the Discussion a detailed explanation of how lysoptosis differs from lysosome-dependent cell death as defined by the Nomenclature Committee on Cell Death. The authors do not entirely understand the comment on Figure 5, as the conclusion based on the data shown is that there is *no evidence* of a role for ferroptosis, pyroptosis, or necroptosis, and only some evidence of caspase-dependent cell death in the HT3 background (with none in the SW756 background). Thus, we have revised the title of this section/figure to read “Cell death in B3-KO cells following RT suggests potential engagement of multiple cell death mechanisms including apoptosis/caspase-dependent cell death and lysosome-dependent necrosis with features of lysoptosis,” in response to the reviewers’ suggestion.